



# Representation of spatial and temporal variability in large-domain hydrological models: Case study for a mesoscale prealpine basin

Lieke Melsen[1], Adriaan Teuling[1], Paul Torfs[1], Massimiliano Zappa[2], Naoki Mizukami[3], Martyn Clark[3], and Remko Uijlenhoet[1]

[1]Hydrology and Quantitative Water Management Group, Wageningen University, Wageningen, The Netherlands.
[2]Swiss Federal Research Institute (WSL), Birmensdorf, Switzerland.
[3]National Centre for Atmospheric Research (NCAR), Boulder, CO, USA.

*Correspondence to:* Lieke Melsen (lieke.melsen@wur.nl)

**Abstract.** The transfer of parameter sets over different temporal and spatial resolutions is common practice in many large-domain hydrological modelling studies. The degree to which parameters are transferable across temporal and spatial resolutions is an indicator for how well spatial and temporal variability are represented in the models. A large degree of transferability may well indicate a poor

representation of such variability in the employed models. To investigate parameter transferability over resolution in time and space we have set-up a study in which the Variable Infiltration Capacity (VIC) model for the Thur basin in Switzerland was run with four different spatial resolutions ($1 \times 1$ km, $5 \times 5$ km, $10 \times 10$ km, lumped) and evaluated for three relevant temporal resolutions (hour, day, month), both applied with uniform and distributed forcing. The model was run 3,150

times using a Hierarchical Latin Hypercube Sample and the best 1% of the runs was selected as behavioural. The overlap in behavioural sets for different spatial and temporal resolutions was used as indicator for parameter transferability. A key result from this study is that the overlap in parameter sets for different spatial resolutions was much larger than for different temporal resolutions, also when the forcing was applied in a distributed fashion. This result suggests that it is easier to

transfer parameters across different spatial resolutions than across different temporal resolutions. However, the result also indicates a substantial underestimation in the spatial variability represented in the hydrological simulations, suggesting that the high spatial transferability may occur because the current generation of large-domain models have an inadequate representation of spatial variability and hydrologic connectivity. The results presented in this paper provide a strong motivation to

further investigate and substantially improve the representation of spatial and temporal variability in large-domain hydrological models.





## 1 Introduction

The history of modern hydrological modelling dates back to halfway the nineteenth century, starting with empirical models to predict peak flows (Todini, 2007). Computational power and increased

data availability have driven the development of increasingly complex and distributed hydrological models (Boyle et al., 2001; Liu and Gupta, 2007). Distributed hydrological models can incorporate spatially varying parameters, including those reflecting land use and soil characteristics (Carpenter and Georgakakos, 2006), and spatially variable forcing. For a long time, hydrological models were developed only at the catchment scale, evolving from empirically-based to more physically-based.

In 1989 the first Global Hydrological Model (GHM) was presented (Vörösmarty et al., 1989; Sood and Smakhtin, 2015). Continuing improvements in computational power and data availability provides new opportunities for GHMs, for example expressed in the recent ambition to develop global models with a resolution in the order of $\sim$ 1 km and higher, the so-called hyper-resolution (Wood et al., 2011; Bierkens et al., 2014; Bierkens, 2015).

Because the parameters in hydrological models often represent a different spatial scale than the observation scale, or because conceptual parameters have no directly measurable physical meaning, calibration of hydrological models is almost always inevitable (Beven, 2012). The increased complexity of hydrological models and the increased application domain has resulted in more complex and time consuming optimization procedures for the model parameters. However, although recent

developments in e.g. satellites and remote sensing can provide spatially distributed data to construct and force models, discharge measurements are still required to calibrate and validate model output. Both to decrease calculation time of the optimization procedure and to be able to apply the model in ungauged or poorly gauged basins and areas, many studies have focused on the transferability of parameter values over time, space, and spatial and temporal resolution (e.g. Wagener and Wheater

(2006); Duan et al. (2006); Troy et al. (2008); Samaniego et al. (2010); Rosero et al. (2010); Kumar et al. (2013); Bennett et al. (2016)). Sometimes it is assumed that parameters are directly transferable, for example by calibrating on a coarser time step than the time step at which the model output will eventually be analysed (e.g. Liu et al. (2013); Costa-Cabral et al. (2013)). Troy et al. (2008) rightly question what the effect is of calibrating at one time step and transferring the parameters to

another time step. Their results suggest that the time step had only limited impact on the calibrated parameters and thus on the monthly runoff ratio. On the other hand, Haddeland et al. (2006) found that modelled moisture fluxes are sensitive to the model time step. Several studies (e.g. Littlewood and Croke (2013); Kavetski et al. (2011); Wang et al. (2009) and Littlewood and Croke (2008)) have found that parameter values are closely related to the employed time step of the model. Chaney et al.

(2015) investigated to what extent monthly runoff observations could reduce the uncertainty around the flow duration curve of daily modelled runoff. They found a decrease in the uncertainty around the flow duration curve when the monthly discharge observations were used, but the magnitude of the reduction was dependent on climate type.





Less intuitive and less common is to transfer parameters across different grid resolutions. Haddeland
et al. (2002) showed that the output of the Variable Infiltration Capacity (VIC) model was signifi-
cantly different when the parameters of the model were kept constant for several spatial resolutions.
For the same model, Liang et al. (2004) showed that model parameters calibrated at a coarse grid
resolution could be applied to finer resolutions to obtain comparable results. Troy et al. (2008) on
the contrary, found that calibrating the VIC model on a coarse resolution significantly affected the
model performance when applied to finer resolutions. Finnerty et al. (1997) investigated parameter
transferability over both space and time for the Sacramento model, and showed that it can lead to
considerable volume errors.

The impact of transferring parameters across spatial and/or temporal resolutions on model perfor-
mance is thus ambiguous, but relevant in the light of hydrological model development, especially for
GHMs which are at the upper boundary of computational power and data availability. Calibration
on a coarse temporal or spatial resolution and subsequently transferring to a higher resolution could
potentially reduce computation time, and it is therefore relevant to investigate the opportunities. But
parameter transferability across spatial and temporal resolutions is also interesting for another rea-
son: it is an indicator for the degree to which spatial and temporal variability are represented in the
model. Ideally, in a model that describes all relevant hydrological processes correctly, parameters
should to a large extent be transferable over time because longer time steps are simply an integration
of the shorter time steps. On the other hand, parameters should not or hardly be transferable over
space, because the physical characteristics which they represent are different from place to place. In-
vestigating parameter transferability across spatial and temporal resolutions can thus provide insight
in the model's representation of spatial and temporal variability.

In this study, we employ the Variable Infiltration Capacity (VIC) model (Liang et al., 1994), which
has also been applied at the global scale (Nijssen et al., 2001; Bierkens et al., 2014), to study param-
eter transferability across temporal and spatial resolutions, accounting for the difference between
uniform and distributed forcing. We applied this study on a well-gauged meso-scale catchment in
Switzerland (the Thur basin, 1703 km$^2$) on spatial resolutions that are relevant for hyper-resolution
studies ($1 \times 1$ km, $5 \times 5$ km and $10 \times 10$ km, as well as a lumped model which represents the $0.5°$ grid
used in many global studies). We use the most common temporal resolutions for which discharge
data are available (hourly, daily, monthly). We ran the models both with distributed forcing (differ-
ent forcing input for each grid cell) and with uniform forcing (same forcing input for each grid cell),
where the latter is in line with many of the datasets currently used for forcing global hydrological
models (e.g. WATCH forcing data, $0.5°$).

Several studies already investigated scale effects in the VIC model, for instance Haddeland et al.
(2002); Liang et al. (2004); Haddeland et al. (2006); Troy et al. (2008); Wenger et al. (2010); Wen
et al. (2012). Novel in this study is that we choose a probabilistic rather than a deterministic ap-
proach: essentially we employ a GLUE-based approach (Beven and Binley, 1992, 2014) in which




we implicitly account for parameter uncertainty. We quantify parameter transferability by evaluating the overlap in behavioural sets for different temporal and spatial resolutions. To determine the behavioural sets, we make use of three different objective functions focusing on high flows, average conditions, and low flows. Novel is also that we test the effect of forcing on the results, and that we use several subbasins to explain the results. Our case study provides a benchmark for parameter transferability for models applied at larger scales, dealing with the same spatial and temporal resolutions as employed here. The results of our study also provide an indication of the current status of spatial and temporal representation in the VIC model, being representative for a larger group of land-surface models.

## 2 Catchment and Data Description

### 2.1 Thur basin

The Thur basin (1703 km$^2$, see Figures 1 and 2) in North-East Switzerland was chosen as study area, because of the excellent data availability in this area and because of its relevanceas a tributary of the river Rhine (Hurkmans et al., 2008). The main river in the basin (the Thur) has a length of 127 km. The average elevation of the basin is 765 m a.s.l., the mean slope is 7.9° (based on a $200 \times 200$ m resolution DEM and slope file). The basin outlet is situated at Andelfingen at an elevation of 356 m a.s.l. (Gurtz et al., 1999). The basin has an alpine/pre-alpine climatic regime, with high temperature variations both in space and time (Figure 3). Precipitation varies from 2500 mm yr$^{-1}$ in the mountains to 1000 mm yr$^{-1}$ in the lower areas. Part of the year the basin is covered with snow. The most striking feature in the Thur basin is the Säntis, an Alpine peak with an altitude of 2502 meter. The dominant land use in the Thur basin is pasture. Within the Thur basin, measurements for nine (nested) sub-catchments are available, see Figure 2. The smallest gauged sub-catchment is the Rietholzbach catchment (3.3 km$^2$, see Seneviratne et al. (2012)), the largest is Halden (1085 km$^2$). Both the Rietholzbach and the Thur have been subject of many previous studies (e.g. Gurtz et al. (1999, 2003); Jasper et al. (2004); Abbaspour et al. (2007); Yang et al. (2007); Teuling et al. (2010); Melsen et al. (2014)). In this study, we will mainly focus on the outlet of the Thur basin.

### 2.2 Discharge data

For the station at the Thur outlet (Andelfingen) and eight sub-basins hourly discharge measurements for the period 1974-2012 were made available by the Swiss Federal Office for the Environment (FOEN). All discharge measurements have been obtained using a stage-discharge relation, based on several measurements conducted by FOEN throughout the years, a.o. with an ADCP. The discharge measurements for the Rietholzbach catchment were made available by ETH Zürich.



### 2.3 Forcing data

Forcing data for this study were made available by the Swiss Federal Office for Meteorology and
Climatology (MeteoSwiss). These data have previously been used for numerous applications of hy-
drological models in the Thur (Jasper et al., 2004; Abbaspour et al., 2007; Fundel and Zappa, 2011;
Fundel et al., 2013; Jörg-Hess et al., 2015). The data are available for this study in the form re-
quired to implement the PREVAH model (Viviroli et al., 2009a, b). Data from nine different me-
teorological stations throughout the catchment (Güttingen, Hörnli, Reckenholz, Säntis, St.Gallen,
Tänikon, Wädenswil, Zürich and Rietholzbach) were available with an hourly time resolution and
spatially interpolated with the use of the WINMET tool of the PREVAH modelling system (Viviroli
et al., 2009a), using elevation-dependent regression (EDR) and inverse distance weighting (IDW)
and combinations of IDW and EDR. The data is available for the period 1981–2004, for which a
stable configuration of stations is available. In this study, we only used data for the period May 2002
- August 2003. To force the VIC model, hourly precipitation, incoming shortwave radiation, tem-
perature, vapour pressure and wind data were used. We have run the model with two set-ups: fed
with uniform forcing and fed with distributed forcing. We ran the model with uniform forcing (equal
input for all the grid cells) in order to isolate the effect of spatially distributed soil parameters. Thus,
the gridded meteorological input obtained with WINMET was averaged for the whole target area.
Because the Thur basin has an extent of approximately $0.5°$, a lumped application of the forcing
mimics the use of global forcing data sets like the WATCH forcing product and the ERA-interim
product. Application with distributed forcing implied different forcing input for each grid cell. Be-
cause of the pronounced elevation differences in the basin, precipitation and temperature show a
clear spatial pattern, which can be seen in Figure 3.

### 150 2.4 Spatial data for the model

Land use, hydraulic conductivity, elevation, and soil water storage capacity maps, all with a spa-
tial resolution of $200 \times 200$ m, were provided by the Swiss Federal Institute for Forest, Snow and
Landscape Research (WSL) under license of Swisstopo (JA100118). Also in this case we used the
pre-processing routines created to implement the PREVAH modelling system (Viviroli et al., 2009a).
The resolution of the available data ($200 \times 200$ m) is higher than the model with the highest resolu-
tion in this study ($1 \times 1$ km), which allows for sub-grid variability in the VIC model for land use and
elevation parameters (see Section 3.1). Other soil characteristics, such as bulk density, have been ob-
tained from the Harmonized World Soil Database (FAO et al., 2012), which has a spatial resolution
of $1 \times 1$ km.





## 3  Model and routing description

The VIC model (version 4.1.2.i) was run at an hourly time step in the energy balance mode, which implies that both the water- and the energy balance are solved. The default routing developed for VIC by Lohmann et al. (1996) is only applicable at daily time steps and hence is not suitable to study parameter transferability at finer temporal resolutions. Therefore, horizontal water transport through

the channel network was implemented using mizuRoute (Mizukami et al., 2015b), developed by the National Centre for Atmospheric Research (NCAR).

### 3.1  The VIC model

The VIC model (Liang et al., 1994, 1996) is a land-surface model that solves the water and the energy balance. Subgrid land use type variability is accounted for by providing vegetation tiles that

each cover a certain percentage of the total surface area. Three different types of evaporation are considered by the VIC model; evaporation from the bare soil ($E_b$), transpiration by the vegetation ($T$), considered per vegetation tile, and evaporation from interception ($E_i$). The total evapotranspiration is the area-weighted sum of the three evaporation types. The fraction of land that is not assigned to a particular land use type is considered to be bare soil. Evaporation from bare soil only occurs

at the top layer (layer 1). If layer 1 is saturated, bare soil evaporation is at its potential rate. Potential evaporation is obtained with the Penman-Monteith equation. If the top layer is not saturated, an Arno-formulation (Francini and Pacciani, 1991), which uses the structure of the Xinanjiang model (Zhao et al., 1980), is used to reduce the evaporation.

For the upper two soil layers, the Xinanjiang formulation (Zhao et al., 1980) is used to describe infil-

tration. This formulation assumes that the infiltration capacity varies within an area. Surface runoff occurs when precipitation added to the soil moisture of layers 1 and 2 exceeds the local infiltration capacity of the soil. Moisture transport from layer 1 to layer 2 and from layer 2 to layer 3 is gravity driven and only dictated by the moisture level of the upper layer. It is assumed that there is no diffusion between the different layers. Layer 3 characterizes long term soil moisture response,

e.g. seasonality. It only responds to short-term rainfall when both top layers are fully saturated. The gravity driven moisture movement is regulated by the Brooks-Corey relationship:

$$Q_{i,i+1} = K_{\text{sat},i} \left( \frac{W_i - W_{r,i}}{W_i^c - W_{r,i}} \right)^{expt_i}. \tag{1}$$

$Q_{i,i+1}$ is the flow [$L^3 T^{-1}$] from layer $i$ to layer $i+1$. $K_{\text{sat},i}$ is the saturated hydraulic conductivity of layer $i$, $W_i$ is the soil moisture content in layer $i$, $W_i^c$ is the maximum soil moisture content in layer

$i$, $W_{r,i}$ the residual moisture content in layer $i$. The exponent of the Brooks-Corey relation, $expt_i$ is defined as follows: $\frac{2}{B_p} + 3$, in which $B_p$ is the pore size distribution index. The exponent as a whole is often calibrated.

Base flow is determined based on the moisture level of layer 3. Base flow generation follows the conceptualization of the Arno model (Francini and Pacciani, 1991). This formulation consists of a





linear part (lower moisture content regions) and a quadratic part (in the higher moisture regions). Baseflow is modelled as follows:

$$Q_b = \begin{cases} \frac{d_s d_m}{w_s W_3^c} \cdot W_3 \text{ if } 0 \leq W_3 \leq w_s W_3^c \\ \\ \frac{d_s d_m}{w_s W_3^c} \cdot W_3 + \left(d_m - \frac{d_s d_m}{w_s}\right)\left(\frac{W_3 - w_s W_3^c}{W_3^c - w_s W_3^c}\right)^g \\ \qquad \text{if } W_3 \geq w_s W_3^c \end{cases}$$

In this equation, $Q_b$ is the total baseflow over the model time step (in this study one hour), $d_m$ is the maximum base flow, $d_s$ the fraction of $d_m$ where non-linear base flow begins, $w_s$ is the fraction of

soil moisture where non-linear baseflow starts. $W_3^c$ is the maximum soil moisture content in layer 3, calculated as a product of porosity and depth. The exponent $g$ is by default set to two (Liang et al., 1996).

Since the grid-size of the VIC model is often larger than the characteristic scale of snow processes, sub-grid variability is accounted for by means of elevation bands. For each grid cell the percentage of

area within certain altitude ranges is provided. The snow model is applied for each elevation range and land use type separately; the weighted average provides the output per grid cell. This output consists of the Snow Water Equivalent (SWE) and the snow depth. The snow model is a two-layer accumulation-ablation model, which solves both the energy- and the mass balance. At the top layer of the snow cover the energy exchange takes place. A zero energy flux boundary is assumed at the

snow-ground interface. A complete description of the model can be found in Liang et al. (1994) and Liang et al. (1996).

### 3.2 Routing

The mizuRoute routine (Mizukami et al., 2015b) takes care of the transport of water between the different grid cells. The routing is based on the same concept as the routing described by Lohmann

et al. (1996), except that in mizuRoute the response is determined per subcatchment (with sizes in the order of 1 km$^2$) instead of per grid cell. Using this approach, the total size of the Thur catchment in the model does not change with the resolution.

With the linearized St. Venant equation,

$$\frac{\partial Q}{\partial t} = D \frac{\partial^2 Q}{\partial x^2} - C \frac{\partial Q}{\partial x}, \tag{2}$$

water is transported from the boundary of the subcatchment to the next subcatchment and finally to the outlet. In Equation 2, $D$ (m$^2$s$^{-1}$) represents the diffusion coefficient and $C$ (m s$^{-1}$) the advection coefficient.

It is important to note that with the applied routing-setup, the drainage network is kept independent of the resolution, because surface runoff is routed for pre-defined sub-basins instead of per grid cell.

In the default VIC routing of Lohmann et al. (1996), water is routed per grid cell and therefore de-



pendent on the spatial resolution of the VIC model. We have excluded the effect of spatial resolution on routing.

## 4 Experimental set-up

We have constructed four VIC models with different spatial resolutions: $1 \times 1$ km, $5 \times 5$ km, $10 \times$ 10 km, as well as a lumped model. These models have been run with both uniform and distributed forcing. Since for the lumped model there is no difference between uniform and distributed forcing, this leads to a total of seven different model set-ups. Because the runtime of the model combined with all the post-processing is rather long (on average 2.5 hours for the $1 \times 1$ km model on a standard PC), an efficient sampling strategy was designed. The procedure we followed is illustrated in Figure 5. With sensitivity analysis (Section 4.4) the most sensitive parameters from the model were selected. Subsequently, we sampled the full parameter space with a uniform prior using a Hierarchical Latin Hypercube sample (HLHS) (Vořechovský, 2015), see Section 4.5. Although sampling the parameter space with a uniform prior is less efficient than other distributions which focus more on the most likely regions, we did not want to exclude any region because both the temporal and spatial resolution were varied. The sampled parameters were applied uniformly over the catchment, although we will test the correctness of this approach by using the information from the subcatchments (see Section 5.3). After running the models with the HLHS, the output was evaluated and the best 1% of the runs was defined as behavioural. The overlap in behavioural sets was used as an indicator for parameter transferability (Section 4.7).

### 4.1 Spatial model resolution

Four VIC implementations with different spatial resolutions ($0.0109°$ roughly corresponding to $1 \times 1$ km, $0.0558° \approx 5 \times 5$ km, $0.1100° \approx 10 \times 10$ km, as well as a lumped model) were constructed. The $1 \times 1$ km model represents the so-called hyper-resolution. Several studies already explore GHMs at this resolution, e.g. Sutanudjaja et al. (2014) for the Rhine-Meuse basin. The model with the $10 \times 10$ km resolution can be characterized as 'regional'. The $5 \times 5$ km model is in between the hyper-resolution scale and the regional scale. The lumped model, which represents an area of 1703 km$^2$, is in the order of magnitude of grid cells with a $0.5°$ resolution, which represents the original scale for which VIC was developed. Figure 1 gives an overview of the cells for the four models. The sampled parameters (see Section 4.4) have been applied uniformly over the catchment, all other parameters have been applied in a distributed manner. We will discuss the effect of applying the sampled parameters uniformly by using data from the nine subcatchments.





### 4.2 Temporal model resolution

The models are run at an hourly time step, implying that they solve both the energy and the water
balance. The hourly output of the routing model is aggregated to daily and monthly time steps for
further evaluation, see Figure 1.

### 4.3 Simulation period

The four models are run for a period of 1 year and four months. The first three months are used
as spin-up period and not used for further analysis. Tests with the same parameter set and different
initial conditions revealed that three months are sufficient to eliminate the effect of initial conditions
(see Figure 4). The initial soil moisture content of the model before spin-up was fixed at $\theta = 0.9$
because we found that the model reaches equilibrium faster when starting from a wet state. The
models have not been subjected to a validation procedure on another time period, because in this
particular application the goal was not to identify the best performing model, but to investigate the
role of temporal and spatial resolution on parameter transferability.

The analysed period is 1 August 2002 – 31 August 2003 (see Figure 4). This period is charac-
terised by three very high peaks (August, September 2002) as well as the severe 2003 drought (June,
July, August 2003). The 2002 peaks (see e.g. Schmocker-Fackel and Naef (2010)) are the highest
peaks measured in the last 39 years (1974-2012) at the outlet of the Thur (right panel in Figure 4).
The peaks were caused by a larger system that also caused the heavy floods in the Elbe and the
Danube (Becker and Grünewald, 2003). In contrast, the 2003 summer was extremely warm and dry
in Western and Central Europe (Miralles et al., 2014), with Switzerland being among the hottest and
driest regions (Andersen et al., 2005; Rebetez et al., 2006; Zappa and Kan, 2007; Seneviratne et al.,
2012).With these two extremes the selected period covers a large part of the flow duration curve,
both in the high and the low flow regions (right panel in Figure 4).

### 4.4 Model parameters

The VIC model has a large number of parameters, divided over three sections: soil parameters,
vegetation parameters, and snow parameters. To determine which parameters should be sampled in
this study, a sensitivity analysis was conducted on a broad selection of parameters. The parameter
selection was made such that the main hydrological processes were represented and included 28
VIC parameters from the three different sections. Sensitivity analysis was conducted using the Dis-
tributed Evaluation of Local Sensitivity Analysis (DELSA) method (Rakovec et al., 2014). DELSA
is a hybrid local-global sensitivity analysis method. It evaluates parameter sensitivity based on the
gradients of the objective function for each individual parameter at several points throughout the pa-
rameter space. Note that this method only provides first-order sensitivities and thus does not account



for parameter interaction.

A base set of 100 parameter samples was created. For each parameter $k$ that is accounted for in the analysis, the base set of parameter samples is perturbed. In total, including the base set, this leads to (number of parameters+1) $\times$ 100 parameter samples that need to be evaluated. To save computation time, the sensitivity analysis was conducted on the lumped VIC model for the Thur. To study the effect of scale on sensitivity, two lumped models for subbasins of the Thur have been constructed: The Jonschwil catchment (495 km$^2$) and the Rietholzbach catchment (3.3 km$^2$). The Rietholzbach catchment is nested inside the Jonschwil catchment, which is again nested in the Thur catchment (Figure 2). The three catchments have comparable land use. The Kling-Gupta Efficiencty of the discharge (KGE(Q)), Nash-Sutcliffe efficiency of the discharge (NSE(Q)) and the Nash-Sutcliffe efficiency of the logarithm of the discharge (NSE(logQ)) (see Section 4.6) were used as objective function to assess the sensitivity of the parameters.

The analysis showed that parameter sensitivity did not notably change over the assessed scales: the same parameters were found to be most sensitive, but in a slightly different order. There are four parameters which, for all scales and for all objective functions, proved to be highly sensitive: The parameter describing variable infiltration ($b_i$), the parameter that defines the fraction of $d_{s,max}$ where non-linear baseflow starts ($d_s$), the maximum velocity of the base flow ($d_m$) and the exponent of the Brooks-Corey relation ($\frac{2}{B_p} + 3$, $expt_2$, see Equation 1). Hence, these four parameters were selected for the sampling analysis. Other parameters that showed sensitivity in some cases were the depth and bulk density of soil layer 2, the depth and bulk density of soil layer 3, and the rooting depth of layer 1. The selection of sensitive parameters closely resembles the results of Demaria et al. (2007), who applied a sensitivity analysis on VIC over different hydroclimatological regimes. Because Demaria et al. (2007) found that the depth of soil layer 2 was highly sensitive, this parameter was added to the selection of parameters that was sampled. The high sensitivity of soil parameters under humid conditions is also in line with studies using other modelling concepts (e.g. Teuling et al. (2009)). In addition, the two routing parameters $C$ and $D$ were sampled because they control the lateral exchange of water between grid cells. An overview of the selected parameters is given in Table 1. In the distributed VIC models (Section 4.1), these parameters have been applied uniformly over the cells.

### 4.5 Hierarchical Latin Hypercube Sample

In comparison with traditional sampling methods, the number of parameter samples needed to cover the full parameter space can decrease significantly by selecting only the most sensitive parameters (see Figure 5b). For the four VIC models (three distributed models, one lumped model) the selected parameters (Table 1) were varied using a Latin Hypercube Sample (LHS). This is a variance reduction method which efficiently samples the parameters within each region with equal probability in the parameter distribution (Vořechovský and Novák, 2009) (see Figure 5c). Especially for the





$1 \times 1$ km model the calculation time is rather long. Therefore, the LHS should preferably be as small as possible, while still being able to provide insights in e.g. posterior parameter distributions. For a Monte Carlo (MC) sample, it is easy to start with a small sample, and add more samples if this shows to be necessary, e.g. based on the sample variance. For a variance reduction technique such as LHS this is not that straight forward. Therefore, we make use of the Hierarchical Latin Hypercube Sample (HLHS), recently developed by Vořechovský (2015). This method allows us to start with a small LHS and add more samples if necessary, while conserving the LHS-structure (Figure 5d). Inherent to this method is that every sample extension is twice as large as the previous sample, which results in a total number of simulations after $r$ extensions:

$$N_{sim,r} = 3^r \cdot N_{start}, \tag{3}$$

with $N_{sim}$ being the total number of simulations, $r$ the number of extensions, and $N_{start}$ the start number of samples. As a starting sample size 350 is chosen, which is sampled based on a space-filling criterion. For the seven parameters in the HLHS sample a uniform prior is assumed in order the study the full parameter space. The starting sample can be increased by a first extension to 1,050 samples in total, further to 3,150, and even up to 9,450. After each extension, the cumulative distribution function (CDF) of the objective functions (KGE, NSE) is compared with the CDF of the previous extension. A Kolmogorov-Smirnov test is used to test if the CDFs are significantly different. It was found that the CDF estimated from 3,150 samples was not significantly different from the CDF based on 1,050 samples at a 0.05-significance level. Therefore, 3,150 samples was considered sufficient to sample the parameter space.

### 4.6 Objective functions

For each model run, several objective functions were evaluated. The three objective functions are:

– The Kling-Gupta Efficiency (KGE) to describe the overall capability of the model to simulate the discharge (Gupta et al., 2009):

$$KGE(Q) = 1 - \sqrt{(r-1)^2 + (\alpha-1)^2 + (\beta-1)^2}, \tag{4}$$

where $r$ is the correlation between observed discharge $Q_o$ and modelled discharge $Q_m$, $\alpha$ is the standard deviation of $Q_m$ divided by the standard deviation of $Q_o$, and $\beta$ is the mean of $Q_m$ ($\overline{Q}_m$) divided by the mean of $Q_o$ ($\overline{Q}_o$).

– The Nash-Sutcliffe Efficiency (NSE) of the discharge to describe the model performance for the higher discharge regions (Nash and Sutcliffe, 1970):

$$NSE(Q) = 1 - \frac{\sum_{t=1}^{T}(Q_o^t - Q_m^t)^2}{\sum_{t=1}^{T}(Q_o^t - \overline{Q}_o)^2} = 2 \cdot \alpha \cdot r - \alpha^2 - \beta_n^2, \tag{5}$$

in which $\beta_n$ is the bias normalized by the standard deviation.



– The Nash-Sutcliffe Efficiency of the logarithm of the discharge NSE(logQ) to test the model
         performance for low discharges (Krause et al., 2005).

The objective functions are calculated for all runs (3,150) for the seven different VIC set-ups and
based on hourly, daily and monthly time steps. Additionally, several other diagnostics have been cal-
culated, among others the Relative Volume Error (RVE) and the autocorrelation, and several flood

and drought characteristics. To characterize floods, the discharge and the timing of the three (one)
highest peaks has been stored for hourly and daily (monthly) time steps. For drought characteriza-
tion, the number of droughts, average drought duration, and the total deficit based on a daily time
step has been obtained (Tallaksen and Van Lanen, 2004). A period was defined as a drought as soon
as discharge was below the 30-days moving average Q90 (Wanders et al., 2015), the lowest 10% of

the discharge based on 39 years of data.

### 4.7    Determination of behavioural sets and parameter transferability

After running the VIC model with 3,150 parameter sets, a selection is made of the best parameter
sets, the so-called behavioural runs (Beven and Binley, 1992). The best 1% (which is different for
different objective functions) of the 3,150 runs (32 members) are selected as behavioural. For each

combination of spatial and temporal resolution, and for the three objective functions separately, the
32 best members are selected. We value all 32 parameter sets equally plausible and do not assign
weights to the best performing sets within the behavioral selection, to account for uncertainty in
the observations. We define parameter transferability $\underset{\longleftrightarrow}{\theta}$ as the percentage agreement in selected
behavioural sets:

$$\underset{\longleftrightarrow}{\theta} = \#(A_{S_i,T_j} \cap B_{S_k,T_l})/n \cdot 100, \tag{6}$$

in which $A_{S_i,T_j}$ is the set of selected behavioural members for spatial resolution $S_i$ and temporal
resolution $T_j$, and $B_{S_k,T_l}$ are the selected members for spatial resolution $S_k$ and temporal resolution
$T_l$. The $n$ is the total number of selected members (in this case 32). Equation 6 expresses $\underset{\longleftrightarrow}{\theta}$ as a
percentage; if $\underset{\longleftrightarrow}{\theta} = 100$, this indicates that for two different resolutions (either spatial, temporal or

both) exactly the same parameter sets were selected as behavioural.

## 5    Results

First, the impact of temporal and spatial resolution on model performance is discussed for both uni-
form and distributed forcing, followed by a discussion of the impact of the temporal and spatial
resolution on parameter distribution. For these analyses, the temporal and spatial resolution are as-

sumed to be independent. Subsequently, the parameter transferability across temporal and spatial
resolution is assessed by determining the overlap in behavioural sets as defined by Equation 6. Af-





ter that, parameter transferability over both temporal and spatial resolution is assessed. Finally, we investigate parameter transferability over the sub-basins of the Thur.

### 5.1 Impact of temporal and spatial resolution on model performance and parameter distribution


Figure 6 shows the model performance of the behavioural sets for the different spatial and temporal resolutions and the different objective functions, both for uniform and distributed forcing. We will first discuss the results for the uniform forcing.

With uniform forcing, the lumped model outperforms the distributed models for all three objective

functions and time steps. The monthly time step shows for all three objective functions an increasing model performance with decreasing spatial resolution. It is remarkable that the model with the monthly time step outperforms the models with daily and hourly time step when the NSE(logQ) was used as objective function, while with the NSE(Q) as objective function exactly the opposite is the case. It is important to notice here that the monthly model results are simply an aggregation from the

hourly model results which might imply that the higher score on the monthly time step is the result of errors which compensate for each other, and that the model perfomance scores for the monthly time step are based on a considerable lower number of points. The KGE(Q) as objective function does not lead to a remarkably different model performance for the monthly time step. From the figure it seems that both the spatial and temporal resolution have impact on the model performance.

This is confirmed with a statistical test. An ANOVA analysis with two factors (temporal resolution; spatial resolution), with three, respectively four levels (hourly, daily, monthly; $1 \times 1$ km, $5 \times 5$ km, $10 \times 10$ km and lumped) shows that both the spatial and the temporal resolution have significant ($p < 0.05$) impact on all three objective functions.

Distributed forcing leads in all cases except one ($1\times1$ km, monthly, NSE(logQ)) to an improved

model performance compared to uniform forcing. It is important to note that for the lumped model uniform and distributed forcing are the same. It should therefore be remarked that while with uniform forcing the lumped model outperforms the other model set-ups, for the distributed forcing the $10\times10$ km model outperforms the other spatial resolutions (except for NSE(logQ)). An ANOVA analysis confirmed that also for distributed forcing, both spatial and temporal resolution have signif-

icant ($p < 0.05$) impact on the model performance for all three objective functions.

Figure 7 shows the distribution of the behavioural sets for the three separate components of the KGE(Q). Regarding the correlation $r$, the monthly time step scores higher than the daily and hourly time step. On the other hand, the hourly and daily time steps score higher with respect to $\beta$ (closer towards 1). Although Figure 6 gives the impression that the model performance in terms of KGE(Q)

is relatively insensitive to temporal and spatial resolution, Figure 7 reveals this is actually the result of compensations from the three different components of the KGE(Q): The monthly time step has a higher correlation, while the daily and hourly time steps have a higher $\beta$.





Figure 8 shows the parameter distribution of the seven sampled parameters, and shows how the distribution varies as a function of temporal and spatial resolution, both for distributed and uniform forcing. The distribution of the behavioural parameter sets for the daily and hourly time steps are very much alike for all parameters, but the distribution for the monthly time step is in some cases broader, which implies that the parameters are less clearly defined. The parameter showing the clearest effect of temporal scale is the advection coefficient $C$ (Figure 8). The $C$ parameter, the velocity component in the routing, becomes less well defined with increasing time step, which is intuitive because timing becomes less relevant for longer time intervals.

The difference in the parameter distribution when comparing distributed and uniform forcing is limited. The clearest difference can be found for the $d_m$-parameter with the NSE(Q) as objective function. This parameter describes the maximum velocity of the base flow, and can potentially impact short term processes for which distributed forcing seems important, like surface runoff. However, there are other parameters, such as the $b_i$-parameter, which are more directly linked to infiltration and surface runoff processes and do not show a clear difference in parameter distribution between distributed and uniform forcing.

With an ANOVA analysis, the significance of temporal and spatial resolution on the parameter distribution of the behavioural sets was tested. Figure 9 shows that the significance of spatial and temporal resolutions on the parameter distribution depends on which objective function was used to determine the behavioural sets. Uniform and distributed forcing show comparable patterns. In general, the temporal resolution has more impact on the parameter distribution (at least four parameters are significantly affected by temporal resolution) than the spatial resolution (only one parameter for one objective function experiences significant impact of the spatial resolution). Only two parameters are significantly impacted by the tempvoral resolution for all three objective functions: $d_s$ and $C$.

## 5.2 Parameter transferability

The main research question of this study is to what extent parameters are transferable across temporal and spatial resolutions, and we will use that as indicator for the representation of spatial and temporal variability in the model. We have defined parameter transferability $\overleftrightarrow{\theta}$ as the percentage agreement in identified behavioural sets (Equation 6). Table 2 and Table 3 give an overview of $\overleftrightarrow{\theta}$ for different temporal and spatial resolutions, both for uniform and distributed forcing. Table 2 shows that the $\overleftrightarrow{\theta}$ is generally high for different spatial resolutions, which suggests that the parameters are to a large extent transferable across spatial scales. In contrast, Table 3 shows that parameters are hardly transferable over the temporal scale. The selected runs for hourly and daily time steps largely agree, but the selected runs on a monthly time step are clearly different. Surprisingly, this is also strongly related to the objective function. The selection based on the NSE(logQ) is less sensitive to temporal resolution than those based on the KGE(Q) or the NSE(Q). A possible explanation is that the NSE(logQ) tends to put more focus on lower discharges with a longer time scale, with less focus



on the short term flashy response of a catchment. Parameter transferability over space is in general

slightly lower when distributed forcing is used compared to uniform forcing. On the other hand, parameter transferability over time is slightly higher for distributed forcing. Decreased sensitivity for the temporal resolution and increased sensitivity for the spatial resolution can indicate an improved physical representation with distributed forcing compared to uniform forcing, as one would expect. Table 2 and 3 list the parameter transferability over only one dimension (either spatial resolution

or temporal resolution). We also investigated the combined effect of transferring parameters over both the spatial and the temporal resolution. Figure 10 shows a linearly fitted field for $\underleftrightarrow{\theta}$ based on KGE(Q) for uniform forcing, fitted through the data points of this study ($R^2 = 0.68$). It clearly shows that temporal resolution has a stronger impact on parameter transferability than spatial resolution. In Figure 10 we assumed a linear model. This is a strong assumption: the terms could be

non-linear and intuitively there could be an interaction term. Due to the limited number of points and the discrete structure of these points, we refrained from developing non-linear and interaction models. The linear regression equation that describes the surface in Figure 10 is given below:

$$\underleftrightarrow{\theta}_{KGE(Q)} = 83.3 - 12.6 \cdot \frac{T_j}{T_l} - 3.0 \cdot \frac{S_i}{S_k}, \tag{7}$$

in which $\frac{T_j}{T_l}$ is the ratio in temporal resolution between the two model set-ups over which parameters

are transferred and $\frac{S_i}{S_k}$ is the ratio in spatial resolution (L/L) between the two model set-ups. The effect of temporal resolution on parameter transferability is stronger (slope of 12.6) than the effect of spatial resolution (slope of 3.0). Parameter transferability decreases when the ratio between the original and the intended spatial and temporal resolution increases. The surfaces based on NSE(Q) ($R^2$=0.60) and NSE(logQ) ($R^2$=0.75) show a similar behaviour:

$$\underleftrightarrow{\theta}_{NSE(Q)} = 88.6 - 12.8 \cdot \frac{T_j}{T_l} - 2.8 \cdot \frac{S_i}{S_k}, \tag{8}$$

$$\underleftrightarrow{\theta}_{NSE(logQ)} = 92.9 - 7.4 \cdot \frac{T_j}{T_l} - 3.6 \cdot \frac{S_i}{S_k}. \tag{9}$$

When we fit a surface through the points obtained for the models run with distributed forcing, the linear regression equations ($R^2$=0.66, 0.67, 0.88 respectively) look as follows:

$$\underleftrightarrow{\theta}_{KGE(Q)} = 80.3 - 11.4 \cdot \frac{T_j}{T_l} - 2.6 \cdot \frac{S_i}{S_k}. \tag{10}$$

$$\underleftrightarrow{\theta}_{NSE(Q)} = 75.3 - 10.3 \cdot \frac{T_j}{T_l} - 4.3 \cdot \frac{S_i}{S_k}, \tag{11}$$

$$\underleftrightarrow{\theta}_{NSE(logQ)} = 91.3 - 5.4 \cdot \frac{T_j}{T_l} - 2.8 \cdot \frac{S_i}{S_k}. \tag{12}$$



Also with distributed forcing, the slope for the temporal resolution is steeper than the slope for spatial resolution, implying that parameter transferability is more sensitive for temporal than for spatial resolution. Compared to uniform forcing, the slope for temporal resolution, and hence the impact of temporal resolution on transferability, is less steep for distributed forcing, while the slope for spatial resolution is on average very comparable for both forcings.


### 5.3   Spatially distributed parameters

The advantage of distributed hydrological models over lumped models is that distributed models can incorporate spatially varying parameters, including those reflecting land use and soil characteristics (Carpenter and Georgakakos, 2006), and spatially varying forcing. Figure 11 for example, shows

how the spatial variation in bulk density decreases with increasing resolution. However, in this study, as in most large-domain studies with distributed models, the most sensitive parameters (i.e. the once that were calibrated) have been applied uniformly over the grid cells. The main motivation for this practice is the ill-posedness of the problem (too many parameters have to be identified with too little information), in addition to computational time. This implies that the advantage of a

distributed model remains unused for the parameters that impact model output most. To test the spatial distribution of the most sensitive parameters for the Thur basin, we have investigated parameter transferability between the Thur basin and the nine subbasins for which discharge data were available (see Section 2.1 and Figure 2). Table 4 gives an overview for a selected number of spatial and temporal resolutions. The table shows that parameter transferability from the Thur to the subbasins is

notably low. An extreme example is the St.Gallen catchment, which has maximum one behavioural parameter set in common with the Thur basin. Table 4 therefore shows that the spatial variation in the calibrated parameters is underestimated in the current model set-up.

### 6   Discussion

### 6.1   Model performance

It seems counter-intuitive that model performance is significantly affected by both the temporal and spatial resolution, while the parameter distribution is mainly impacted by the temporal resolution. This can be explained, however. Model performance can still be significantly impacted by temporal and spatial resolution, even if the same parameters are selected for different spatial resolutions. This implies that the model performance is mainly limited by the model structure or set-up, and much

less by the parameter values. This is confirmed by comparing the uniform and distributed forcing. Although the distribution of the behavioural parameters was not very different for the two forcing types, the model performance for distributed forcing was in almost all cases better than the model performance for the uniform forcing.



Liang et al. (2004) defined a so-called 'critical resolution', beyond which a finer spatial resolution

would not lead to any improvement in the model performance. In the study of Liang et al. (2004) this critical resolution for the VIC model was found to be $1/8°(\approx 12.5 \times 12.5$ km). All spatial resolutions applied in this study but the lumped one are below this critical resolution. The results in this study are therefore consistent with the results from Liang et al. (2004), because we did not find any improvement in model performance with increasing spatial resolution, neither for the uniform nor

for the distributed forcing. Rather, we find the contrary; for the uniform forcing the lumped model outperformed the higher resolution models, and for the distributed forcing the $10 \times 10$ km outperformed the other models. If something like a critical resolution exists, it is probably related to the processes represented in the model. Contradictory to our findings are the results of Zappa (2002), who found that a critical spatial resolution in the Thur region is in the order of $500 \times 500$ m using the

PREVAH model, because of the complex topography and snow processes in the catchment. This can either imply that the sub-grid variability parametrization in VIC is effective, or that not all relevant hydrological processes are included in the VIC model. In order to check this last suggestion, future research on parameter transferability should consider more hydrological fluxes and states besides the discharge, e.g. evapotranspiration.

**6.2   The high sensitivity for temporal resolution**

The conclusion that parameters cannot be transferred across temporal resolution seems to contradict the results of Troy et al. (2008). The large difference is that Troy et al. (2008) only used sub-daily time steps (1, 3, 6, 12 hours), whereas we did find agreement between the hourly and daily time step. Therefore, our results are not necessarily contradictory. Troy et al. (2008) chose the sub-daily

time steps in order to investigate if time could be saved in the calibration process by calibrating on a coarser time step. Unfortunately, the reality is that in most large-domain studies models are calibrated with monthly discharge observations (Melsen et al., 2015) rather than with sub-daily observations. Our results suggest that models which were calibrated or validated at a monthly time step cannot be interpreted at the daily or hourly time step. Chaney et al. (2015) showed that monthly

discharge observations could decrease the uncertainty around the daily flow duration curve. The decrease in uncertainty by adding monthly discharge information differed for different climates. The Thur basin, with a wet continental climate, would experience a high reduction in uncertainty. This means that our results, which show that with monthly data it is impossible to determine the optimal parameter set for the hourly or daily time step, would even be stronger for dry climates (Chaney

et al., 2015). Kavetski et al. (2011) showed that parameter values can significantly change by changing the temporal resolution. They found that the sensitivity of a parameter to temporal resolution could be related to the model structure; the parameters from simpler model structures were more sensitive to temporal resolution than the parameters from more complex models.

Figure 12 shows that the conclusions we draw from Table 2 and Table 3 are not only valid for the



best 1% of runs selected as behavioural. This figure gives an overview for two selected cases, which
show that model performance deteriorates when parameters are transferred over time, also for the
best 10% up to higher thresholds, whereas the impact of spatial resolution on model performance
deterioration is limited.

### 6.3 Models versus nature: Do the current generation of models adequately represent spatial
variability?

Our results show that parameter transferability is more sensitive to temporal than for spatial reso-
lution. A key question is to what extent this result stems from the model representation of spatial
variability. Spatial variability can be reflected in three domains of the model: the forcing, the rout-
ing, and the soil- and land use parameters (of which some are calibrated). In this study we excluded
the effect of routing by using a high-resolution drainage network independent of the resolution of
the hydrologic model. We investigated the effect of forcing by comparing the results for distributed
and uniformly applied forcing, we aggregated soil- and land use parameters for lower resolutions
(Figure 11), and we tested parameter transferability from the Thur to subbasins to estimate spatial
variation in the calibrated parameters. Despite distributed forcing and the decrease in variation in
soil- and land use parameters, the model parameters showed low sensitivity to the spatial resolution.
A possible explanation could be the sub-grid parametrizations of the VIC model for land use and
elevation, which decrease the effect of up-scaling these parameters to other resolutions, as shown
by Haddeland et al. (2002). However, we think that Section 5.3 and Table 4 also show how spatial
variability is underestimated by calibrating and applying the most sensitive parameters uniformly
over the basin.

The models in this study are configured in a similar way to many current day large-domain hydrolog-
ical models, using common data like the Harmonized World Soil Database and uniform application
of the most sensitive parameters. As such, this study is likely representative for many large-domain
studies. The limited sensitivity for spatial resolution is arguable because our implementation of VIC
substantially underestimates the spatial variability in nature, and, importantly, that similar issues in
representing spatial variability is a common problem in large-domain hydrological modelling (e.g.,
see the model configuration in Mizukami et al. (2015a)). Many studies have considered spatial vari-
ability in forcing (Adams et al., 2012; Lobligeois et al., 2014) and soil parameters (Mohanty and
Skaggs, 2001; Western et al., 2004). Kim et al. (1997) accounted for heterogeneity in soil hydraulic
properties using stochastic methods, based on the scaling theory of Miller and Miller (1956). In
fact, the effect of stochastic soil parametrizations on parameter transferability would be a valuable
research topic (Maxwell and Kollet, 2008). We argue here that the high spatial transferability may
occur because the current generation of land-surface models have an inadequate representation of
spatial variability and hydrologic connectivity, providing a strong motivation to substantially im-
prove the representation of spatial and temporal variability in models. This not only implies increas-



ing the spatial (and temporal) resolution of the model, but also including more relevant hydrological processes.

### 6.4 Limitations of this case study

The results in our study are based on a limited number of model configurations for a single basin, so the results presented here are only intended to provide an example of the behaviour in the current generation of land-surface models. Our results show a low sensitivity for the spatial resolution, whether applied with distributed forcing or not. The observed impact of spatial resolution can therefore almost completely be attributed to the effect of spatially distributed soil and land use parameters (including the calibrated ones), which could be substantially underestimated. The impact of temporal resolution on parameter transferability is large. We employed the temporal resolutions for which most hydrological observations are available, thus our results are relevant for practical applications. Based on the work of Chaney et al. (2015) we expect that parameter transferability will be lower for arid climates than the numbers we obtained, and based on the work of Kavetski et al. (2011) we expect that parameter transferability will be lower for more parsimonious models. The general message from our study is the surprisingly high spatial transferability, highlighting the need for a focused research effort to improve the representation of spatial variability in large-domain distributed models (GHMs). A possible path forward is to develop computationally frugal process representations, as for example presented by Hazenberg et al. (2015) for hillslope processes.

### 7 Summary and conclusions

A VIC model for the Thur basin was run with four different spatial resolutions ($1 \times 1$ km, $5 \times 5$ km, $10 \times 10$ km, lumped) and evaluated at three different temporal resolutions (hourly, daily, monthly). The forcing was applied both uniformly and distributed over the catchment, and the drainage network was defined independent of the model resolution. Three objective functions were used to evaluate model performance: KGE(Q), NSE(Q) and the NSE(logQ). The model was run 3,150 times using a Hierarchical Latin Hypercube Sample and the best 1% of the runs was selected as behavioural and used for further analysis. Parameter transferability was quantified by evaluating the overlap in behavioural sets for different temporal and spatial resolutions. From the results we can draw the following conclusions:

- Both the spatial resolution and the temporal resolution of the VIC model had a significant impact on the model performance, either expressed in terms of KGE(Q), NSE(Q), or NSE(logQ). The model performance evaluated at a monthly time step consistently increased with decreasing spatial resolution, while for the daily and hourly time step no clear relation with spatial resolution could be found. Generally, the models applied with spatially distributed forcing performed better than the models applied with uniform forcing.



– The spatial resolution of the model had little impact on the parameter distribution of the be-
        havioural sets. On the other hand, the temporal resolution significantly impacted the distribu-
        tion of at least four out of seven parameters, both for uniformly and distributed forcing.

        – Parameters could to a large extent be transferred across the spatial resolutions, while parameter
        transferability over the temporal resolutions was less trivial. Parameter transferability between
the hourly and daily time step was found to be feasible, but the monthly time step lead to
        substantially different parameter values. This is crucial information, because many studies
        tend to calibrate the VIC model on the monthly time step (Melsen et al., 2015). The results
        of this study suggest that the output from models calibrated on a monthly time step cannot be
        interpreted or analysed on a daily or hourly time step. This might seem obvious, but it should
be recognized that the increasing spatial resolution of large-domain land-surface models might
        increase the expectations concerning temporal resolution as well, as described in Melsen et al.
        (2015).

        – We also investigated if parameters could be transferred across both the spatial and the tempo-
        ral resolution simultaneously. Parameter transferability decreases when the ratio between the
original and the intended spatial and/or temporal resolution increases. The ratio of temporal
        resolutions has a larger negative effect on parameter transferability than the ratio of spatial res-
        olutions. It was also shown that parameter transferability depends on the objective function.
        When the NSE(logQ), which tends to put more emphasize on low flows, is used as evaluation
        criterion, the parameter values at a monthly time step overlap much more with the daily and
hourly time steps than when KGE(Q) or NSE(Q) are used as objective functions. This means
        that parameter transferability across temporal resolution also depends on the time scale of the
        process to which a particular parameter refers.

The most important result of our study is that it showed high parameter transferability across spa-
tial resolution, even when forcing was applied in a distributed fashion. A possible explanation for
the low sensitivity to spatial resolution is the uniform application of the most sensitive parameters.
This is indicative of a substantial underestimation of the actual spatial variability represented by
the VIC simulations. We did, however, construct our model according to current day standards for
large-domain land-surface models, raising the point that the high spatial transferability may occur
because the current generation of models have an inadequate representation of spatial variability
and hydrologic connectivity. The results presented in this paper provide strong motivation to further
investigate and substantially improve the representation of spatial and temporal variability in large-
domain hydrological models.

Large-domain hydrological models have many applications, from water footprints (Gleeson et al.,
2012) and water scarcity (Hoekstra, 2014), to global water use (Wada and Bierkens, 2014) and elec-
tricity supply (Van Vliet et al., 2012), but the spatial variability in the models is very likely underes-





timated, which increases the uncertainty in the model results. A critical evaluation of large-domain hydrological models on a smaller scale, as was done in this study, shows that we should be carefull with interpreting the results of large-domain models.

*Acknowledgements.* The authors would like to thank Kevin Sampson for the preparation of GIS files for the
routing, Oldrich Rakovec for providing and helping with DELSA, and Miroslav Vořechovský for the provided Hierarchical Latin Hypercube Sample. The Swiss Federal Office for the Environment (FOEN) and Martin Hirschi and Dominic Michel from ETH Zürich are thanked for kindly providing the discharge data. We would like to thank MeteoSwiss for providing the forcing data. Lieke Melsen would like to acknowledge Niko Wanders, Wouter Greuell, Pablo Mendoza, Rohini Kumar, Stephan Tober and Oldrich Rakovec for fruitful discus-
sions that led to the basis of this paper. The data in this study are available from the first author upon request.





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





**Table 1.** Sampled model parameters.

| Parameter | Units | Lower value | Upper value | Description |
| --- | --- | --- | --- | --- |
| $b_i$ | - | $10^{-5}$ | 0.4 | Variable infiltration curve parameter |
| $d_s$ | - | $10^{-4}$ | 1.0 | Fraction of $d_{s,max}$ where non-linear baseflow starts |
| $d_m$ | mm d$^{-1}$ | 1.0 | 50 | Maximum velocity of the baseflow |
| $expt_2$ | - | 4.0 | 18.0 | Exponent of the Brooks-Corey drainage equation for layer 2 |
| $Depth_2$ | m | $Depth_1$+0.1 | $Depth_1$+3 | Depth of soil layer 2 |
| $C$ | ms$^{-1}$ | 0.5 | 4 | Advection coefficient of horizontal routing (St. Venant) |
| $D$ | m$^2$s$^{-1}$ | 200 | 4000 | Diffusion coefficient of horizontal routing (St. Venant) |

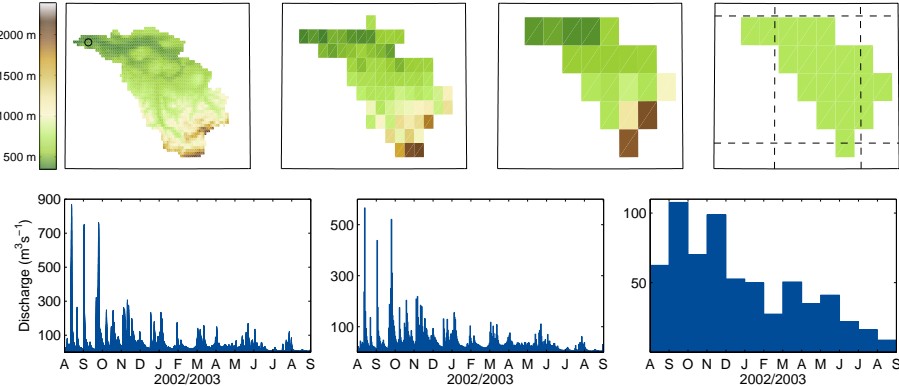

**Figure 1.** Overview of the spatial and temporal resolutions employed in this study. Top from left to right: DEM grid cells for $1 \times 1$ km, $5 \times 5$ km, $10 \times 10$ km resolution and the lumped model. The circle in the left panel shows the location of the Thur outlet where the discharge is measured. The dotted lines in the right panel indicate a $0.5°$ grid. Bottom: The three temporal resolutions, observed discharge at an hourly, daily and monthly time step.





**Table 2.** Transferability of parameters across spatial resolution, expressed as percentage agreement in detected behavioural runs for different spatial resolutions (in km) at different time steps.

| | Uniform forcing (% agreement) | | | Distributed forcing (% agreement) | | |
|---|---|---|---|---|---|---|
| | **HOUR** | | | | | |
| | KGE(Q) | NSE(Q) | NSE(logQ) | KGE(Q) | NSE(Q) | NSE(logQ) |
| $1 \times 1$ vs $5 \times 5$ | 78 | 84 | 91 | 88 | 75 | 84 |
| $1 \times 1$ vs $10 \times 10$ | 72 | 81 | 81 | 78 | 56 | 78 |
| $5 \times 5$ vs $10 \times 10$ | 94 | 94 | 91 | 88 | 81 | 94 |
| $1 \times 1$ vs lumped | 78 | 88 | 91 | | | |
| $5 \times 5$ vs lumped | 91 | 84 | 94 | | | |
| $10 \times 10$ vs lumped | 88 | 81 | 88 | | | |
| | **DAY** | | | | | |
| | KGE(Q) | NSE(Q) | NSE(logQ) | KGE(Q) | NSE(Q) | NSE(logQ) |
| $1 \times 1$ vs $5 \times 5$ | 94 | 84 | 84 | 91 | 84 | 91 |
| $1 \times 1$ vs $10 \times 10$ | 84 | 69 | 69 | 78 | 69 | 81 |
| $5 \times 5$ vs $10 \times 10$ | 91 | 84 | 84 | 89 | 84 | 91 |
| $1 \times 1$ vs lumped | 91 | 81 | 88 | | | |
| $5 \times 5$ vs lumped | 91 | 88 | 94 | | | |
| $10 \times 10$ vs lumped | 84 | 84 | 81 | | | |
| | **MONTH** | | | | | |
| | KGE(Q) | NSE(Q) | NSE(logQ) | KGE(Q) | NSE(Q) | NSE(logQ) |
| $1 \times 1$ vs $5 \times 5$ | 75 | 88 | 88 | 84 | 84 | 91 |
| $1 \times 1$ vs $10 \times 10$ | 66 | 84 | 81 | 66 | 78 | 84 |
| $5 \times 5$ vs $10 \times 10$ | 88 | 91 | 94 | 78 | 88 | 94 |
| $1 \times 1$ vs lumped | 78 | 72 | 94 | | | |
| $5 \times 5$ vs lumped | 78 | 75 | 88 | | | |
| $10 \times 10$ vs lumped | 78 | 78 | 88 | | | |





**Table 3.** Transferability of parameters across temporal resolution, expressed as percentage agreement in detected behavioural runs for different temporal resolutions at different spatial resolutions.

| | Uniform forcing (% agreement) | | | Distributed forcing (% agreement) | | |
|---|---|---|---|---|---|---|
| | \multicolumn 1 × 1 km | | | | | |
| | KGE(Q) | NSE(Q) | NSE(logQ) | KGE(Q) | NSE(Q) | NSE(logQ) |
| hour vs day | 56 | 81 | 81 | 69 | 63 | 75 |
| hour vs month | 3 | 6 | 34 | 6 | 9 | 47 |
| day vs month | 3 | 6 | 47 | 6 | 13 | 63 |
| | 5 × 5 km | | | | | |
| | KGE(Q) | NSE(Q) | NSE(logQ) | KGE(Q) | NSE(Q) | NSE(logQ) |
| hour vs day | 66 | 88 | 81 | 69 | 69 | 81 |
| hour vs month | 3 | 6 | 38 | 9 | 6 | 53 |
| day vs month | 3 | 6 | 47 | 9 | 6 | 66 |
| | 10 × 10 km | | | | | |
| | KGE(Q) | NSE(Q) | NSE(logQ) | KGE(Q) | NSE(Q) | NSE(logQ) |
| hour vs day | 63 | 75 | 78 | 59 | 72 | 78 |
| hour vs month | 3 | 3 | 44 | 13 | 6 | 59 |
| day vs month | 0 | 6 | 63 | 13 | 6 | 75 |
| | lumped | | | | | |
| | KGE(Q) | NSE(Q) | NSE(logQ) | KGE(Q) | NSE(Q) | NSE(logQ) |
| hour vs day | 66 | 84 | 81 | | | |
| hour vs month | 3 | 0 | 44 | | | |
| day vs month | 3 | 3 | 53 | | | |





**Table 4.** Transferability of parameters from the Thur to the nine subbasins, expressed as percentage agreement (%) in detected behavioural runs. The forcing was applied uniformly and the KGE(Q) was used as objective function.

| Catchment (size) | $1 \times 1$ km | | | $5 \times 5$ km | $10 \times 10$ km |
|---|---|---|---|---|---|
| | hour | day | month | hour | hour |
| Rietholzbach (3.3 km$^2$) | 19 | 0 | 0 | 25 | 19 |
| Herisau (17.8 km$^2$) | 16 | 6 | 0 | 16 | 16 |
| Appenzell (74.2 km$^2$) | 28 | 25 | 9 | 28 | 16 |
| Wängi (78.9 km$^2$) | 9 | 56 | 31 | 34 | 50 |
| Mogelsberg (88.2 km$^2$) | 28 | 38 | 66 | 19 | 28 |
| Frauenfeld (212 km$^2$) | 3 | 3 | 75 | 3 | 0 |
| St.Gallen (261 km$^2$) | 3 | 0 | 0 | 3 | 0 |
| Jonschwil (493 km$^2$) | 6 | 0 | 0 | 6 | 0 |
| Halden (1085 km$^2$) | 19 | 9 | 0 | 18 | 13 |

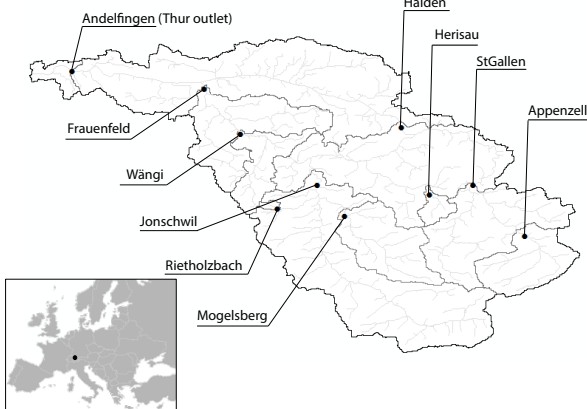

**Figure 2.** The Thur basin and the nine sub-basins for which discharge data were available.




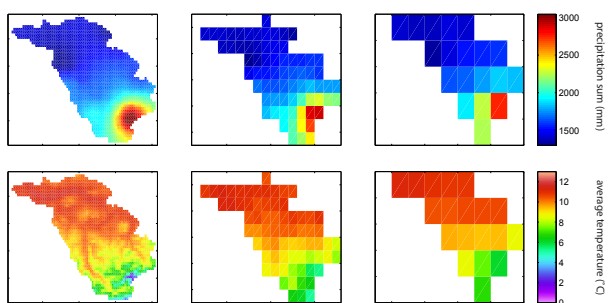

**Figure 3.** Upper panels: The precipitation sum in the Thur catchment over the full model period (1/8/2002 – 31/8/2003) shown for different resolutions (f.l.t.r. $1 \times 1$ km, $5 \times 5$ km, $10 \times 10$ km). Lower panels: the average temperature over this period for the same spatial resolutions.

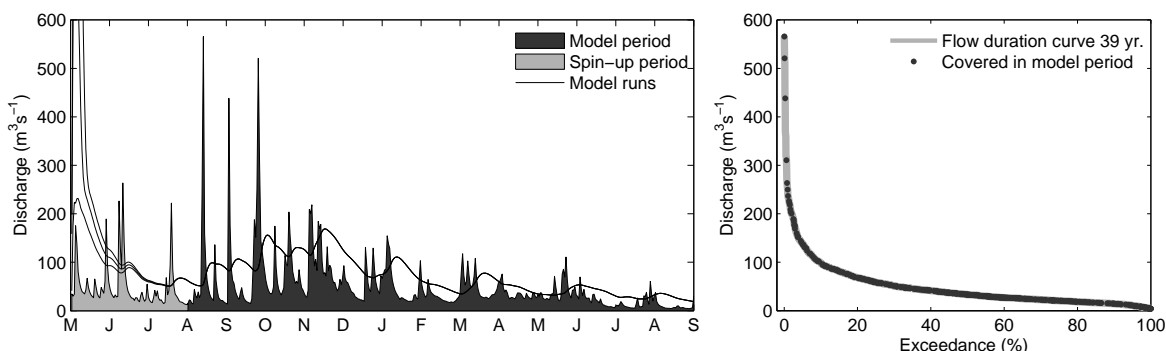

**Figure 4.** Daily discharge characteristics for the Thur basin. Left panel: the daily discharge in the Thur for the selected model period. The black lines show three model runs with the same parameter set but with different initial conditions ($\theta = 0.5, 0.7, 0.9$). Right panel: part of the flow duration curve covered within the model period. The flow duration curve is based on 39 years of daily discharge observations in the Thur for the period 1974–2012.





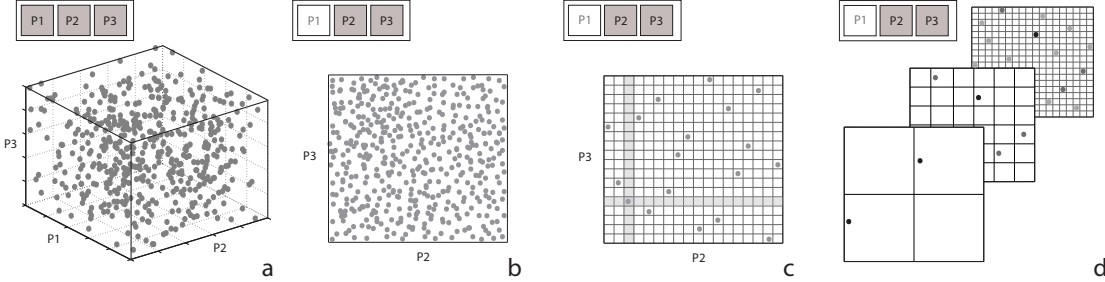

**Figure 5.** Parameter sampling as applied in this study. (a) Example situation when sampling for a model with three parameters. (b) Sensitivity analysis can be conducted to decrease the dimensions of the sampling space. (c) Latin Hypercube sampling is structured and more efficient: one sample in each row and each column, as indicated with the bands. The number of samples has to be determined beforehand. (d) Hierarchical Latin Hypercube sampling allows to extend the sample if necessary, while conserving Latin Hypercube structure.

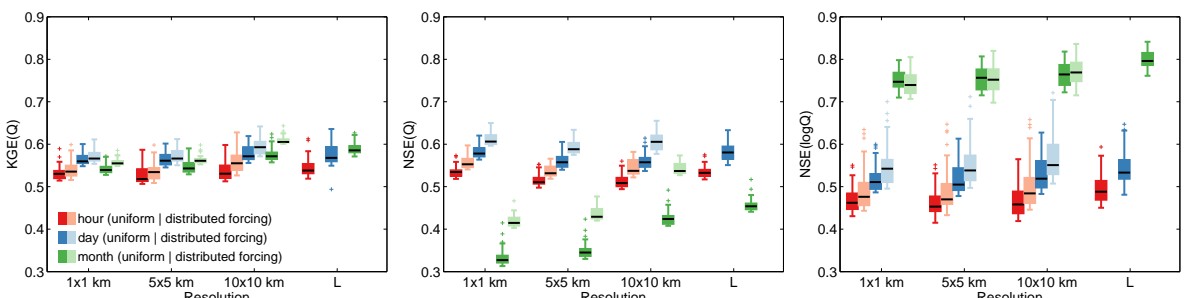

**Figure 6.** Model performance of the behavioural sets for different temporal resolutions and different spatial resolutions. The left panel shows the KGE(Q), the middle panel the NSE(Q) and the right panel the NSE(logQ). Per objective function the most behavioural sets were selected, hence the selected sets where not necessarily the same for the three objective functions. The box shows the 25–75% quantile.





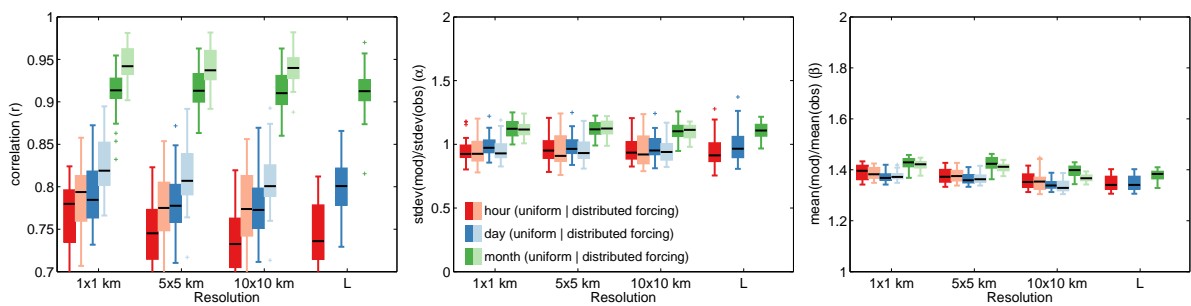

**Figure 7.** The model performance for the three separate components of the Kling-Gupta Efficiency of the behavioural sets for different temporal and spatial resolutions. The left panel shows the correlation $r$, the middle panel the standard deviation of the model output divided by the standard deviation of the observations ($\alpha$), and the right panel shows the mean of the model output divided by the mean of the observations ($\beta$).





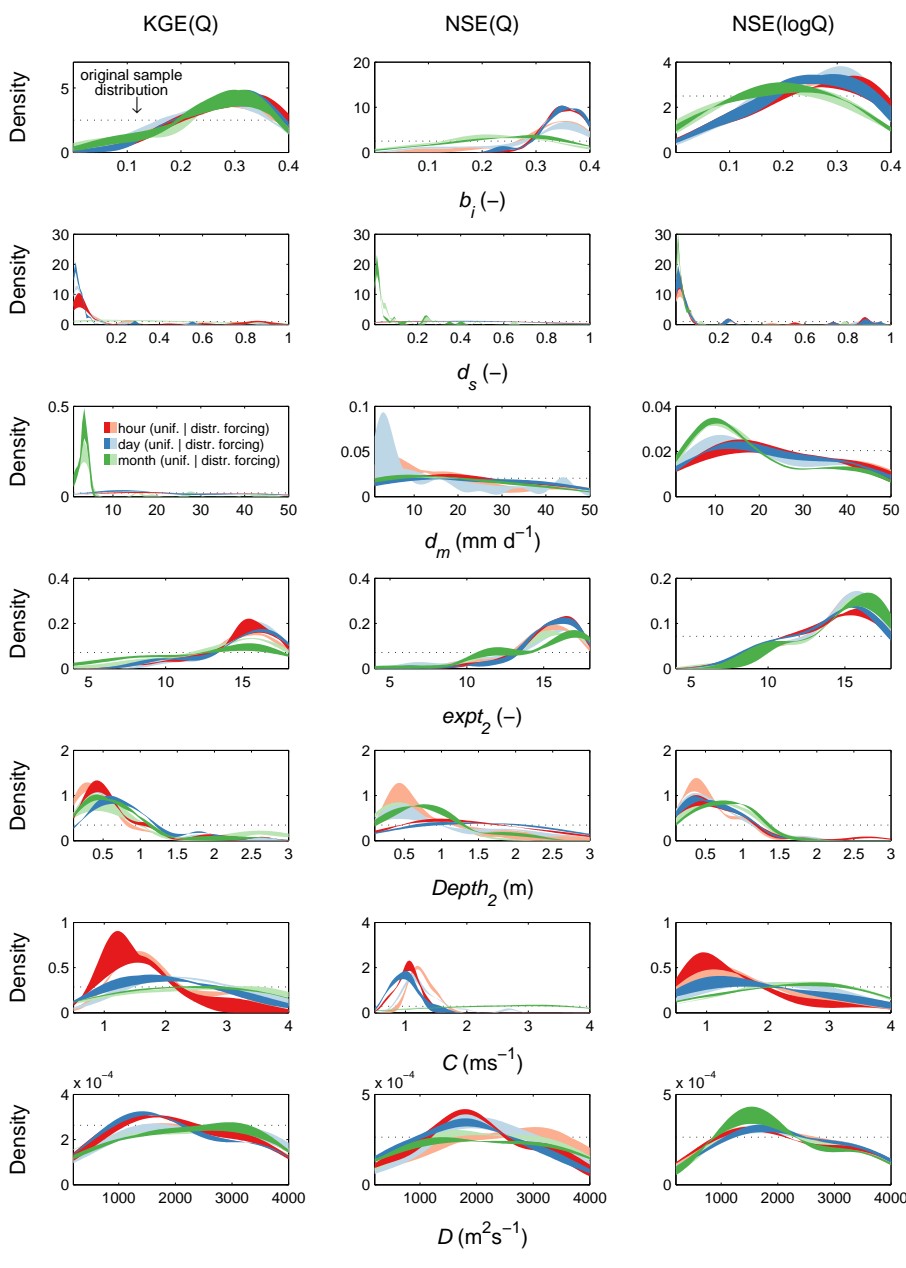

**Figure 8.** Distribution of the sampled parameters for the behavioural sets, fitted with a kernel-density. The width of the line indicates the variation in distribution between the different spatial resolutions. The left column is based on KGE(Q), the middle column on NSE(Q) and the right column on NSE(logQ). Legend according to Figure 6.





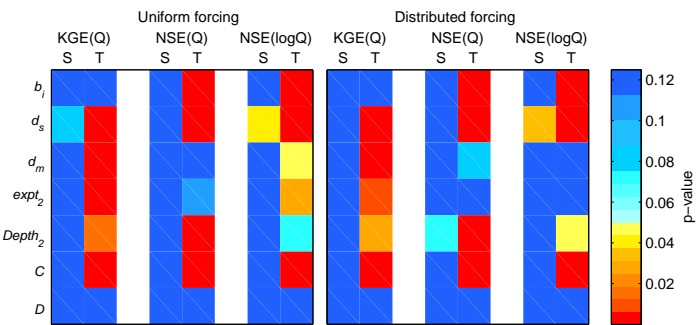

**Figure 9.** The effect of spatial and temporal resolution on parameter distribution. The p-value indicates the significance of the impact of spatial resolution (S) and temporal resolution (T) on the parameter values of the behavioural sets, evaluated for the three objective functions.

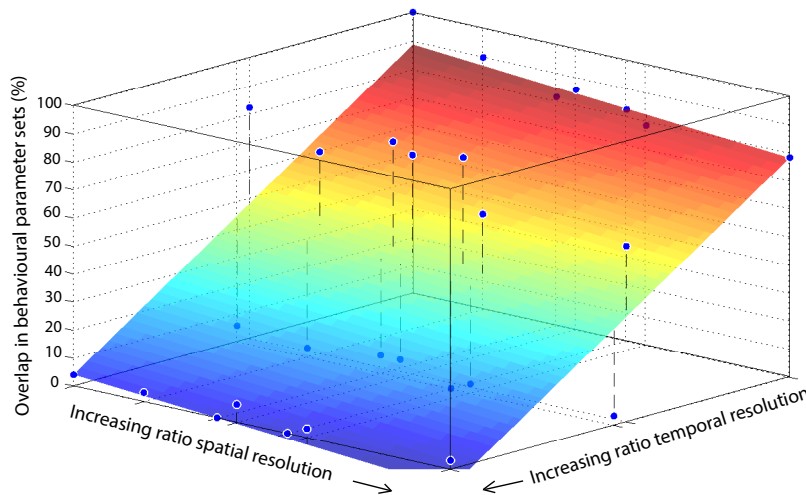

**Figure 10.** Linearly fitted surface describing parameter transferability (Eq. 6) versus ratio in temporal and spatial resolution. The fitted surface has an $R^2$ of 0.68 (Eq. 7). Ratio of temporal resolutions is defined as follows: transfer from hourly to daily time step is a ratio of 24, whereas transfer from hourly to monthly is a ratio of 732 (732 hours in one month of 30.5 days). The ratio of spatial resolutions is defined as the square root of the number of cells that would fit in the other cell: from $1 \times 1$ km resolution to $5 \times 5$ km resolution is a ratio of $\sqrt{25} = 5$. The behavioural sets were determined based on the KGE(Q).





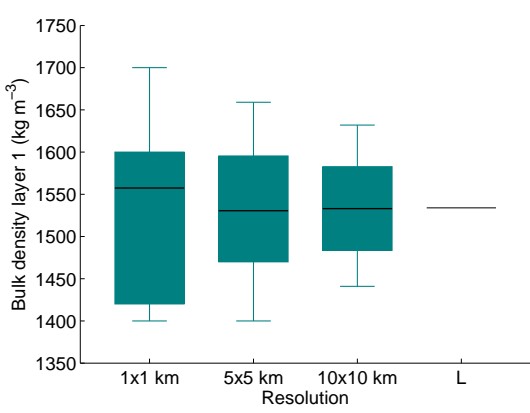

**Figure 11.** Distribution of bulk density over the grid cells for the four different spatial resolutions.





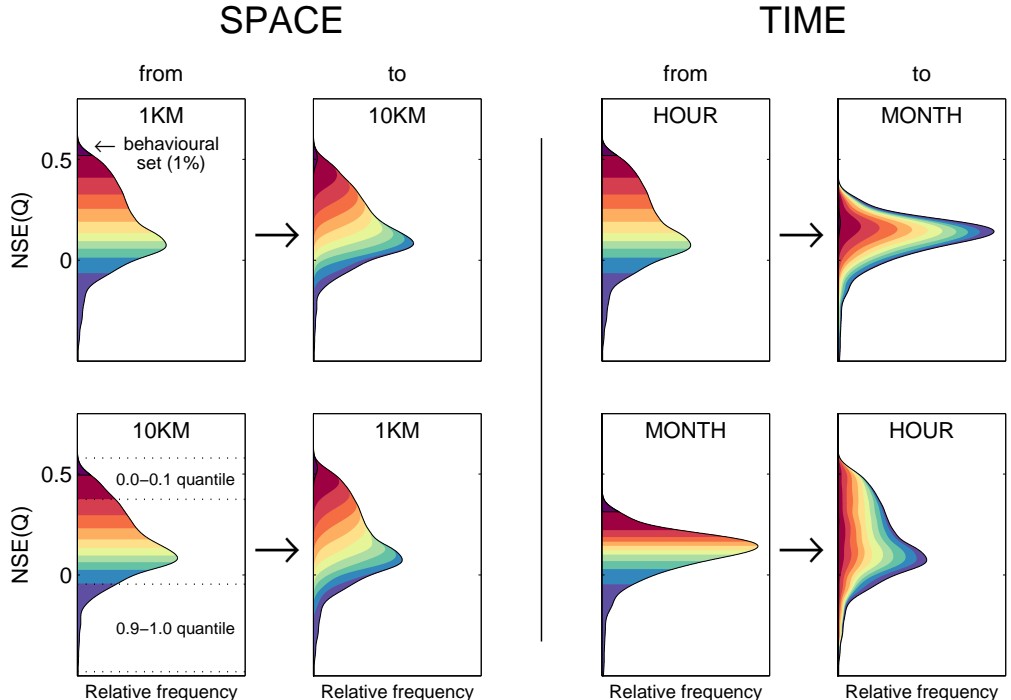

**Figure 12.** Impact of parameter transfer on model performance. The panels show the distribution of the NSE(Q) fitted with a kernel density for 3,150 runs. On the left hand side of the arrow the red area represents the best 10% of the runs, each colour interval increasing with 10% to the full data set (100%, purple). The selected behavioural runs are indicated separately with a black line (best 1%)). The panel on the right hand side of the arrow shows the distribution of the model performance for the coloured selections when evaluated at another spatial (left) or temporal (right) resolution. The data for the first two columns are based on hourly discharges, the data for the second two columns are based on the $1 \times 1$ km model.