# Peer review of "Representation of spatial and temporal variability in large-domain hydrological models: Case study for a mesoscale prealpine basin"

_Hydrology and Earth System Sciences, 2015_

## Referee Comment (RC1) · Anonymous Referee #1 · 14 Feb 2016

Review of "Representation of spatial and temporal variability in large-domain hydrological models: Case study for a mesoscale prealpine basin" by Melsen et al., HESS 2016

This study focuses on the transferability of parameter sets over different temporal and spatial resolutions, using the VIC model on a basin in Switzerland as a case study. The authors use a hierarchical latin hypercube sample to identify behavioral parameter sets at different temporal (hourly, daily, monthly) and spatial (1km, 5km, 10km) resolutions. The study considers the overlap in behavioral parameter sets between different resolutions to be an indicator of transferability, which may indicate a poor model representation of either temporal or spatial variability.
[Figure]

The study is well-designed and executed. The results are presented clearly and address a relevant practical need, as large-scale hydrologic models are often calibrated at a coarser scale than that at which they are later applied, out of concern for computation time. The paper will be a strong contribution to HESS, subject to the clarification of a few minor issues.

(1) The percentage of overlapping behavioral parameter sets across temporal and spatial scales is used as a measure of transferability. The number of behavioral parameter sets in each case is 32. Are there sample size issues here? Should confidence intervals for proportions be reported for the transferability metric? Understandably, the amount of computation time is limited by the most finely resolved scale, so this is not to say that the authors should perform additional model runs. But some discussion of sample size seems warranted. Because the behavioral cutoff is 1% of samples, it might be interesting to see what the results look like with a more lenient restriction (2-5%) to allow a larger sample size in the estimate of the proportion.

(2) Near L95 the GLUE approach is mentioned, i.e., comparing the uninformative prior and behavioral posterior distributions of parameters. Would it have been possible to use this comparison to derive the transferability metric? For example the distance between PDFs or CDFs. It's not clear that this would have been "better" than a proportion metric, but perhaps a bit less reliant on the particular parameter sets that were sampled. Again this is only a matter for clarification.

(3) One methodological point that deserves more explanation is the use of spatially lumped parameters, even when spatial resolutions are increased. Thus the use of different spatial resolutions is really only a matter of distributed forcing data, not the parameter fields themselves. This is explored to some extent in Section 5.3 with the transfer of parameters to sub-basins, which proves difficult. The authors mention on L520 that spatial resolution does not affect the distribution of behavioral parameters very much. This makes sense as distributed forcing (especially considering the heterogeneity of precipitation over the model period as shown in Figure 3) will allow hopefully

for a more accurate time of concentration. The authors claim that spatial variability is underestimated, but how much of this finding is due to the fact that parameters are lumped? The setup and results of the spatial transferability experiment seem to contradict the hypothesis near L80 that parameters should hardly be transferable over spatial scales.

(4) Before the HLHS sample is performed, the authors find that parameter sensitivity (using the DELSA method) does not change much across scales. Is this similar or different to the finding that behavioral parameter sets (values) DO change across scales? Is there an interpretation of this result that can be discussed? Many readers may find parameter sensitivity, and its transferability across scales, equally interesting as the model performance itself.

(5) According to Figure 6, the behavioral parameter sets at finer temporal resolutions (hourly, daily) are not so good at reproducing observed streamflow (NSE $\sim$ 0.5-0.6). This may warrant further discussion. The selection of behavioral parameter sets is based on the top 1% of model runs, not the performance metrics like NSE, KGE, etc. But using those criteria, it may be that none of the model runs are "behavioral". Are there any implications of this?

Again these are largely clarifying points related to the experimental setup. Overall the experiment is well-designed and the paper well-written, with very nice figures to complement the text, and it will make a nice contribution to HESS.

---

## Author Comment (AC1) · 22 Feb 2016

We would like to thank the reviewer for the positive response on our manuscript. Here below we respond to the clarifications that were asked.

(1) Indeed the results could depend on the – subjective – choice to select the best 1% of the runs. This is why we have shown the effect of the complete sample in Figure 12. This Figure shows that the results are consistent among larger sample sizes. We do however agree that this could be discussed in more detail, and therefore we propose to add a discussion with a small sensitivity analysis on 2-5% sample size. This can be summarized in a figure in auxiliary material.

[Figure]

(2) This is an interesting suggestion; we actually did not investigate whether the distance between PDFs or CDFs could be informative for the degree of transferability. We could imagine that some information could be extracted from the agreement or disagreement in PDF or CDF, but we cannot directly think of a way to quantify the transferability based on this approach. In that sense, our current set-up is very clear and straight forward, although some subjectivity is involved in the size of the sample (as discussed under point 1).

(3) This is indeed the most critical point in this study. We would like to elucidate that indeed, the most sensitive parameters, i.e. the parameters that have been sampled, have been applied uniformly over the catchment, but all the other parameters (the soil parameters, land-use parameters, snow parameters) have been applied distributed. Some of these parameters, for example the bulk density of layer 2 and 3, did show relatively high sensitivity in our sensitivity analysis. Therefore, the models with higher spatial resolution could benefit from the distribution of these parameters. But indeed; our conclusion that spatial variability is underestimated is mainly the result of the uniformly applied most sensitive parameters, as shown in Section 5.3. This is, however, part of our conclusion and recommendation. As pointed out in L. 662 we constructed our model according to current day practices; many large scale models apply the calibrated parameters uniformly, there are even examples where fixed parameter values are used for a complete climate zone (see for example Nijssen et al., 2001, "Predicting the Discharge of Global Rivers"). We think that if we want to move towards higher spatial resolution models, eventually hyperresolution, we need to account for the poor representation of spatial variability before higher spatial resolution is of any added value. We will try to formulate this more clearly in the conclusion.

(4) The finding that parameter sensitivity did not change very much across scales is different from the finding that parameter values did not change very much across scales, since equal sensitivity for a certain parameter does not necessarily imply that the value of the parameter is the same. We do have the results of the sensitivity analysis and a

table with all the parameters that have been subject to the sensitivity analysis. We can discuss this in auxiliary material.

(5) It is indeed an effect of our method that, by choosing a fixed percentage rather than a minimum performance, not all the selected runs can or might be considered 'behavioral'. We think that the implications of this effect for our conclusions are limited; it does not necessarily negatively nor positively impact the transferability of the parameters across spatial or temporal resolutions. We will add this discussion to the manuscript. A small note we would like to make is that in a literature review that we performed during this study (see also http://www.hydrol-earth-syst-sci-discuss.net/hess-2015-513/) , we found that for large-scale studies NSE in the order of 0.4 are sometimes already considered as behavioral. . .

We think that, with the suggestions provided by the reviewer, we can increase the readability of the manuscript and make the conclusions clearer.
* * *

---

## Referee Comment (RC2) · Anonymous Referee #2 · 9 Apr 2016

This article compares parameters estimated for the conceptually-based VIC model at varying spatial and temporal resolution over a regional catchment. I find this manuscript to be well written and clearly organized. I think the topic is of interest to the readership of HESS, I recommend publication pending some minor comments. I have listed a couple detailed comments below.

Comments

This study uses a conceptually based approach where calibration is generally mandatory given the indirect nature of the model parameters. Are there lessons learned from the more physically based modeling community which is also studying the effects of resolution (see references below)? It would be valuable to summarize these and dis-

cuss in context with current study.

I am curious about the spatial resolution of routing. It appears that the routing network is a constant across all simulations, which might substantially influence the conclusions. Prior studies have relaxed that assumption (again see references below). The authors should comment on this more and while it may be infeasible to conduct additional simulations, additional discussion would be valuable. I wonder if this might help explain why the authors found good spatial parameter transferability? I also wonder if findings that Dunnian runoff averages to effective parameters (e.g. Meyerhoff in references below) while Hortonian does not might be relevant here as well?

References:

Meyerhoff Quantifying the effects of subsurface heterogeneity on hillslope runoff using a stochastic approach. Hydrogeology 2011

Sciuto et al. Influence of soil heterogeneity and spatial discretization on catchment water balance modeling, VZJ 2010

Shrestha et al. Impacts of grid resolution on surface energy fluxes simulated with an integrated surface-groundwater flow model HESS 2015

Sulis et al. Impact of grid resolution on the integrated and distributed response of a coupled surface-subsurface hydrological model for the des Anglais catchment, Quebec HyP 2011

Vivoni et al. On the effects of triangulated terrain resolution on distributed hydrologic model response HyP 2005

von Gunten et al Efficient calibration of a distributed pde-based hydrological model using grid coarsening JoH 2014

---

## Author Comment (AC2) · 9 Apr 2016

I would like to thank the reviewer for the positive feedback and the provided suggestions. Below a short response is given to the suggestions.

Point 1, lessons from physically based models. I would like to note that sometimes the VIC model is also considered 'physically based', the definition of 'physically based' and 'conceptual' are sometimes diffuse. We will definitely go through the provided literature to compare the approach.

Point 2, the spatial resolution of the routing. The river network has been determined based on sub basins with sizes in the order of ∼1 km. The river network is kept constant

for the different spatial resolutions of the VIC model, i.e.; for the 10x10 km resolution, several sub basins received the same input to rout, whilst for the 1x1 km resolution nearly all sub basins had their 'own' grid cell in the VIC model (see Lines 223-227 in the manuscript). We indeed think that the effect of spatial variation can be increased by adapting the routing scheme. However, this effect is then caused by the routing, and not by the VIC model. For clarity, we excluded the effect of routing (Line 575), but I agree with the reviewer that we could include some discussion on the effect of routing.

---

## Author Response (AR1)

**Rebuttal concerning manuscript hess-2015-532:**
**"Representation of spatial and temporal variability in large-domain hydrological models: Case study for a mesoscale prealpine basin"**

Dear Professor Weiler,

We would like to thank you for the review process and the editor report. Here below you can find our response to the suggestions of you and the reviewers.

- Suggestions from the editor:

**Highlight higher values in Table 2 and 3.**
We have colored the cells in Table 2 and Table 3 based on their value, this indeed helps in recognizing patterns. We have extended the coloring to Table 4 as well.

**The temporal resolution does not seem to show a linear relationship in Figure 10.**
We agree that a more advanced relationship (e.g. quadratic) might be more valid in this case. However, we think that the number of points is too limited to apply more complex models than a linear model, we see figure 10 merely as an illustrative figure to show the strong effect of temporal resolution on transferability as compared to spatial resolution. We have stressed the illustrative character of this figure in the text in Lines 470-473:

*"Figure 10 shows the relative impact of temporal and spatial resolution on parameter transferability based on KGE(Q) for uniform forcing. To illustrate the relative impact of changes in spatial and temporal resolution, we fitted a linear surface through the data points from our study ($R^2$= 0.68). "*

It has also been stressed in the caption of Figure 10: *"The linear surface ($R^2$ = 0.68) was fitted to illustrate the relative impact of changes in spatial and temporal resolution."*

**Please add a color legend for Figure 12.**
A color legend has been added for Figure 12.

- Suggestions from reviewer 1:

**Because the behavioral cutoff is 1% of samples, it might be interesting to see what the results look like with a more lenient restriction (2-5%) to allow a larger sample size in the estimate of the proportion.**
This is shown in the Supplementary Material Table 1 and Table 2 for the KGE(Q). Results are slightly different for this more lenient restriction, but the pattern remains the same, as is also shown in Figure 12. We have added a sentence to refer to the Supplementary Material in Lines 566-569:
*"Figure 12 and Table 2 and 3 in the supplementary material shows that the conclusions we draw from Table 2 and Table 3 are not only valid for the best 1% of runs selected as behavioural. Table 2 and 3 in the supplementary material show that the same patterns are found when selecting the best 2% respectively 5% of the model runs."*

**Near L95 the GLUE approach is mentioned, i.e., comparing the uninformative prior and behavioral posterior distributions of parameters. Would it have been possible to use this comparison to derive the transferability metric? For example the distance between PDFs or CDFs.**
This is an interesting suggestion; we actually did not investigate whether the distance between PDFs or CDFs could be informative for the degree of transferability. We could imagine that some

information could be extracted from the agreement or disagreement in PDF or CDF, but we cannot directly think of a way to quantify the transferability based on this approach. In that sense, our current set-up is very clear and straight forward, although some subjectivity is involved in the size of the sample.

One methodological point that deserves more explanation is the use of spatially lumped parameters, even when spatial resolutions are increased. Thus the use of different spatial resolutions is really only a matter of distributed forcing data, not the parameter fields themselves.
This is indeed the most critical point in this study. We would like to elucidate that indeed, the most sensitive parameters, i.e. the parameters that have been sampled, have been applied uniformly over the catchment, but all the other parameters (the soil parameters, land-use parameters, snow parameters) have been applied distributed. Some of these parameters, for example the bulk density of layer 2 and 3, did show relatively high sensitivity in our sensitivity analysis. Therefore, the models with higher spatial resolution could benefit from the distribution of these parameters. We have stressed this by adding Line 245: *"The sampled parameters were applied uniformly over the catchment, 245 whereas all other soil- and landuse parameters have been applied in a distributed fashion. "*
But indeed; our conclusion that spatial variability is underestimated is mainly the result of the uniformly applied most sensitive parameters, as shown in Section 5.3. This is, however, part of our conclusion and recommendation. We have added Lines 607-609 to further discuss the point of spatially lumped parameters: *"Promising techniques have been developed to allow spatial distribution of calibrated parameters, for example with Multiscale Parameter Regionalization (MPR, Samaniego et al. (2010); Kumar et al. (2013)), which could and should be applied for large-domain hydrologic models."*
Furthermore, we discuss this choice now in lines 320-323: *"Because sampling the seven selected parameters in a distributed fashion is computationally extremely demanding and currently not yet feasible, the sampled parameters have been applied uniformly over the cells in the distributed VIC models. This is according to current practice in large-scale modelling."*

Before the HLHS sample is performed, the authors find that parameter sensitivity (using the DELSA method) does not change much across scales. Is this similar or different to the finding that behavioral parameter sets (values) DO change across scales? Is there an interpretation of this result that can be discussed? Many readers may find parameter sensitivity, and its transferability across scales, equally interesting as the model performance itself.
The finding that parameter sensitivity did not change very much across scales is different from the finding that parameter values did not change very much across scales, since equal sensitivity for a certain parameter does not necessarily imply that the value of the parameter is the same.
In order to provide more insight in the sensitivity analysis, we have added the table with investigated parameters and a figure with the results of the sensitivity analysis to the Supplementary Material. Line 289 and Line 309 have been added to the manuscript to refer to the Supplementary Material.

According to Figure 6, the behavioral parameter sets at finer temporal resolutions (hourly, daily) are not so good at reproducing observed streamflow (NSE ~ 0.5-0.6). This may warrant further discussion. The selection of behavioral parameter sets is based on the top 1% of model runs, not the performance metrics like NSE, KGE, etc. But using those criteria, it may be that none of the model runs are "behavioral". Are there any implications of this?
It is indeed an effect of our method that, by choosing a fixed percentage rather than a minimum performance, not all the selected runs can or might be considered 'behavioral'. We think that the implications of this effect for our conclusions are limited; it does not necessarily negatively nor positively impact the transferability of the parameters across spatial or temporal resolutions. Lines 374-377 have been added to the manuscript:

*"Inherent to our approach, selecting a certain percentage of runs rather than applying a threshold level based on an objective function, is that the selected runs do not necessarily comply to an acceptable model performance. We expect that this neither positively nor negatively influence our results concerning parameter transferability. "*

- Suggestions from reviewer 2:

This study uses a conceptually based approach where calibration is generally mandatory given the indirect nature of the model parameters. Are there lessons learned from the more physically based modeling community, which is also studying the effects of resolution?
We think this is a valuable suggestion, and have added a small paragraph to discuss the effect of spatial resolution in more physically-based models which are usually applied to a smaller area but with a higher resolution, see Lines 70-75: *"Although the ambition of GHMs is to move towards hyperresolution ($\sim 1$ km and higher), more physically-based catchment models have already been applied at spatial resolutions in the order of 100 meters. Also for these models at this scale, the effect of spatial resolution has been investigated (e.g. Vivoni et al. (2005); Sulis et al. (2011); Shrestha et al. (2015)). Even for fully coupled surface- groundwater land-surface models, the effect of spatial resolution on hydrologic fluxes was found to be considerable (Shrestha et al., 2015). "*

I am curious about the spatial resolution of routing. It appears that the routing network is a constant across all simulations, which might substantially influence the conclusions. Prior studies have relaxed that assumption. The authors should comment on this more and while it may be infeasible to conduct additional simulations, additional discussion would be valuable.
It was a conscious choice to exclude the effect of spatial scale on the routing. A discussion on this has been added in Lines 574-581: *"In this study we excluded the effect of routing by using a high-resolution drainage network based on sub-basins with a size of ~1 km$^2$, independent of the resolution of the hydrologic model. We think that the effect of spatial resolution can be increased by adapting the routing scheme accordingly. Drainage network resolution may affect the projected hydrograph, for example with changes in the stream network and the channel slope. However, this effect should then be assigned to the routing model, and not to the runoff generation model (the hydrologic model). For clarity, we decided to exclude the effect of spatial resolution on routing in this study. "*

Besides the suggestions provided by the editor and the reviewers, we have added a relevant recent reference (Ficchi et al., 2016). Furthermore, we have adapted Figure 2 to fit better with the projection of the Thur basin as used in the other figures of the manuscript.

We believe that the suggestions from the reviewers and the editor have improved the manuscript. We hope we have addressed all requests sufficiently.

Kind regards,

Lieke Melsen and co-authors.

[revised manuscript text omitted]

**1 Supplementary material**

[Figure]

**Figure 1.** DELSA parameter sensitivity (scaled from 0 to 1) for three nested basins with a different size (Rietholzbach; 3.3 km$^2$, Jonschwil; 493 km$^2$, Thur; 1703 km$^2$). The numbers on the x-axis refer to the parameters in Table 1. The sensivity as shown in this figure is based on the NSE(Q) as objective function. Results are shown based on a daily and hourly time interval.

**Table 1.** Description and boundary values of parameters that have been considered in the DELSA sensitivity analysis.

| Nr. | Parameter | Units | Lower value | Upper value | Description |
|---|---|---|---|---|---|
| **Soil parameter file** | | | | | |
| 1 | $b_i$ | - | $10^{-5}$ | 0.4 | Variable infiltration curve parameter |
| 2 | Ds | - | $10^{-4}$ | 1 | Fraction of Dsmax where non-linear baseflow starts |
| 3 | Dsmax | mm d$^{-1}$ | 1 | 50 | Maximum velocity of the baseflow |
| 4 | Ws | - | 0.5 | 1 | Fraction of maximum soil moisture where non-linear baseflow starts |
| 5 | c | - | 1 | 4 | Exponent used in the baseflow curve |
| 6 | expt1 | - | 5 | 30 | Exponent of the Brooks-Corey drainage equation layer 1 |
| 7 | expt2 | - | 5 | 30 | Exponent of the Brooks-Corey drainage equation layer 2 |
| 8 | expt3 | - | 5 | 30 | Exponent of the Brooks-Corey drainage equation layer 3 |
| 9 | Ksat1 | mm d$^{-1}$ | 100 | 1000 | Saturated hydrologic conductivity layer 1 |
| 10 | Ksat2 | mm d$^{-1}$ | 100 | 1000 | Saturated hydrologic conductivity layer 2 |
| 11 | Ksat3 | mm d$^{-1}$ | 100 | 1000 | Saturated hydrologic conductivity layer 3 |
| 12 | $Depth_1$ | m | 0.01 | 0.5 | Thickness of soil layer 1 |
| 13$^\dagger$ | $Depth_2$ | m | $Depth_1$+0.1 | $Depth_1$+4 | Thickness of soil layer 2 |
| 14 | $Depth_3$ | m | 0.1 | 4 | Thickness of soil layer 3 |
| 15 | bulk density1 | kg m$^{-3}$ | 1500 | 2685 | Bulk density of soil layer 1 |
| 16 | bulk density2 | kg m$^{-3}$ | 1500 | 2685 | Bulk density of soil layer 2 |
| 17 | bulk density3 | kg m$^{-3}$ | 1500 | 2685 | Bulk density of soil layer 3 |
| 18 | Wcr-FRACT1 | - | 0.30 | 0.47 | Fractional soil moisture content at critical point layer 1 |
| 19 | Wcr-FRACT2 | - | 0.30 | 0.47 | Fractional soil moisture content at critical point layer 2 |
| 20 | Wcr-FRACT3 | - | 0.30 | 0.47 | Fractional soil moisture content at critical point layer 3 |
| 21 | snow-rough | m | $5 \cdot 10^{-5}$ | 0.5 | Surface roughness of the snow pack |
| **Vegetation parameter file** | | | | | |
| 22 | Root depth 1 | m | 0.1 | 3 | Root zone thickness layer 1 |
| 23 | Root depth 2 | m | 0.1 | 3 | Root zone thickness layer 2 |
| 24 | Root depth 3 | m | 0.1 | 3 | Root zone thickness layer 3 |
| **Vegetation library file** | | | | | |
| 25 | rmin | s m$^{-1}$ | 30 | 300 | Minimum stomatal resistance of vegetation |
| 26$^\star$ | LAI | - | 0.7 | 1.3 | Leaf Area Index |
| **Global parameter file** | | | | | |
| 27 | $T_{min}$ | °C | -1.5 | 0.0 | Minimum temperature at which rain can fall |
| 28$^\dagger$ | $T_{max}$ | °C | $T_{min}$+0.5 | $T_{min}$+1.5 | Maximum temperature at which snow can fall |

$^\dagger$ Value of this parameter must be greater than the related parameter mentioned in the parameter boundaries.

$^\star$ Implemented as a multiplication factor to the default parameter values.

**Table 2.** Transferability of parameters across spatial resolution, expressed as percentage agreement in detected behavioural runs based on KGE(Q). The results are shown for three different sample sizes for the behavioural runs; the highest 1% of the runs, the highest 2% of the runs, and the highest 5% of the runs.

| | Uniform forcing (% agreement) | | | Distributed forcing (% agreement) | | |
|---|---|---|---|---|---|---|
| | **HOUR** | | | | | |
| | 1% | 2% | 5% | 1% | 2% | 5% |
| $1 \times 1$ vs $5 \times 5$ | 78 | 89 | 85 | 88 | 84 | 89 |
| $1 \times 1$ vs $10 \times 10$ | 72 | 77 | 78 | 78 | 70 | 83 |
| $5 \times 5$ vs $10 \times 10$ | 94 | 83 | 92 | 88 | 86 | 91 |
| $1 \times 1$ vs lumped | 78 | 88 | 85 | | | |
| $5 \times 5$ vs lumped | 91 | 89 | 92 | | | |
| $10 \times 10$ vs lumped | 88 | 84 | 90 | | | |
| | **DAY** | | | | | |
| | 1% | 2% | 5% | 1% | 2% | 5% |
| $1 \times 1$ vs $5 \times 5$ | 94 | 84 | 86 | 91 | 86 | 89 |
| $1 \times 1$ vs $10 \times 10$ | 84 | 78 | 79 | 78 | 84 | 81 |
| $5 \times 5$ vs $10 \times 10$ | 91 | 91 | 92 | 89 | 94 | 88 |
| $1 \times 1$ vs lumped | 91 | 86 | 87 | | | |
| $5 \times 5$ vs lumped | 91 | 89 | 90 | | | |
| $10 \times 10$ vs lumped | 84 | 84 | 90 | | | |
| | **MONTH** | | | | | |
| | 1% | 2% | 5% | 1% | 2% | 5% |
| $1 \times 1$ vs $5 \times 5$ | 75 | 86 | 85 | 84 | 89 | 86 |
| $1 \times 1$ vs $10 \times 10$ | 66 | 69 | 73 | 66 | 69 | 74 |
| $5 \times 5$ vs $10 \times 10$ | 88 | 83 | 86 | 78 | 73 | 83 |
| $1 \times 1$ vs lumped | 78 | 70 | 70 | | | |
| $5 \times 5$ vs lumped | 78 | 75 | 77 | | | |
| $10 \times 10$ vs lumped | 78 | 72 | 79 | | | |

**Table 3.** Transferability of parameters across temporal resolution, expressed as percentage agreement in detected behavioural runs based on KGE(Q). The results are shown for three different sample sizes for the behavioural runs; the highest 1% of the runs, the highest 2% of the runs, and the highest 5% of the runs.

| | Uniform forcing (% agreement) | | | Distributed forcing (% agreement) | | |
|---|---|---|---|---|---|---|
| | $1 \times 1$ km | | | | | |
| | 1% | 2% | 5% | 1% | 2% | 5% |
| hour vs day | 56 | 77 | 83 | 69 | 72 | 78 |
| hour vs month | 3 | 3 | 6 | 6 | 16 | 15 |
| day vs month | 3 | 3 | 4 | 6 | 9 | 17 |
| | $5 \times 5$ km | | | | | |
| | 1% | 2% | 5% | 1% | 2% | 5% |
| hour vs day | 66 | 70 | 80 | 69 | 73 | 79 |
| hour vs month | 3 | 2 | 8 | 9 | 13 | 15 |
| day vs month | 3 | 3 | 6 | 9 | 9 | 22 |
| | $10 \times 10$ km | | | | | |
| | 1% | 2% | 5% | 1% | 2% | 5% |
| hour vs day | 63 | 73 | 79 | 59 | 84 | 79 |
| hour vs month | 3 | 3 | 8 | 13 | 16 | 16 |
| day vs month | 0 | 3 | 8 | 13 | 16 | 24 |
| | lumped | | | | | |
| | 1% | 2% | 5% | 1% | 2% | 5% |
| hour vs day | 66 | 72 | 82 | | | |
| hour vs month | 3 | 6 | 9 | | | |
| day vs month | 3 | 3 | 8 | | | |